# Neuromorphic object localization using resistive memories and ultrasonic transducers

Filippo Moro [1✉], Emmanuel Hardy [1], Bruno Fain[1], Thomas Dalgaty [1,2], Paul Clémençon [1,3], Alessio De Prà [1,4], Eduardo Esmanhotto[1], Niccolò Castellani[1], François Blard[1], François Gardien[1], Thomas Mesquida[2], François Rummens[2], David Esseni[4], Jérôme Casas[3], Giacomo Indiveri [5], Melika Payvand [5] & Elisa Vianello [1✉]

Real-world sensory-processing applications require compact, low-latency, and low-power computing systems. Enabled by their in-memory event-driven computing abilities, hybrid memristive-Complementary Metal-Oxide Semiconductor neuromorphic architectures provide an ideal hardware substrate for such tasks. To demonstrate the full potential of such systems, we propose and experimentally demonstrate an end-to-end sensory processing solution for a real-world object localization application. Drawing inspiration from the barn owl's neuroanatomy, we developed a bio-inspired, event-driven object localization system that couples state-of-the-art piezoelectric micromachined ultrasound transducer sensors to a neuromorphic resistive memories-based computational map. We present measurement results from the fabricated system comprising resistive memories-based coincidence detectors, delay line circuits, and a full-custom ultrasound sensor. We use these experimental results to calibrate our system-level simulations. These simulations are then used to estimate the angular resolution and energy efficiency of the object localization model. The results reveal the potential of our approach, evaluated in orders of magnitude greater energy efficiency than a microcontroller performing the same task.

[1] CEA, LETI, Université Grenoble Alpes, 38054 Grenoble, France. [2] CEA, LIST, Université Grenoble Alpes, 38054 Grenoble, France. [3] Insect Biology Research Institute, Université de Tours, 37020 Tours, France. [4] DPIA, Università degli Studi di Udine, 33100 Udine, Italy. [5] Institute for Neuroinformatics, University of Zürich and ETH Zürich, 8057 Zürich, Switzerland. ✉email: filippo.moro@cea.fr; elisa.vianello@cea.fr

We are entering an era of pervasive computing, where an exponentially increasing number of devices and systems are being deployed to assist us in our daily lives. These systems are expected to operate continuously, dissipating the lowest possible amount of energy, while learning to interpret the data they capture from several sensors in real time and produce a binary output as the outcome of a classification or recognition task. One of the most important steps required to reach this objective is to extract useful and compact information from noisy and often incomplete sensory data[1]. Conventional engineering approaches typically sample the sensed signals at constant and high rates, thus generating huge amounts of data, even in the absence of useful input stimuli. Moreover, these approaches use sophisticated digital signal processing techniques to pre-process the input (often noisy) data. Conversely, biology provides alternative solutions for processing noisy sensory data, using energy-efficient, asynchronous, event (spike)-driven methods[2,3]. Neuromorphic computing draws inspiration from biological systems to reduce the computational cost in terms of energy and memory requirements, relative to conventional signal processing techniques[4–6]. Innovative general-purpose brain-inspired systems that implement spiking neural networks (TrueNorth[7], BrainScaleS[8], DYNAP-SE[9], Loihi[10], Spinnaker[11]) have recently been demonstrated. These processors offer low-power and low-latency solutions for implementing machine learning tasks, and for modeling cortical circuits. To take full advantage of their energy efficiency, these neuromorphic processors should be connected directly to event-driven sensors[12,13]. However, only few sensory devices exist today that directly provide event-driven data. Prominent examples are the dynamic vision sensor (DVS) used for vision applications such as tracking and motion detection[14–17], the silicon cochlea[18] and neuromorphic auditory sensor (NAS)[19], used for processing auditory signals, the olfactory sensor[20], and multiple examples of touch sensors used for texture recognition[21,22].

In this article, we present a newly developed event-driven auditory processing system applied to object localization. Here, for the first time, we describe an end-to-end system for object localization that is obtained by coupling the state-of-the-art piezoelectric micro-machined ultrasound transducers (pMUTs) to a neuromorphic resistive memory (RRAM)-based computational map. In-memory computing architectures employing RRAMs are a promising solution to reduce energy consumption[23–29]. Their inherent non-volatility—not requiring active power consumption to store or refresh the information— matches the asynchronous event-driven nature of neuromorphic computation perfectly, resulting in virtually no power consumption when the system is idle. Piezoelectric micromachined ultrasound transducers (pMUTs) are low-cost, miniaturized silicon-based ultrasound sensors able to act as emitters and receivers[30–34]. To process the signals captured by the embedded sensors, we have taken inspiration from the neuroanatomy of the barn owl[35–37]. The barn owl *Tyto alba* is known for its exceptional night hunting capabilities made possible by a very efficient auditory localization system. To calculate the position of a prey, the barn owl's localization system encodes the time-of-flight (ToF) of the sound wave coming from the prey when it reaches each of the owl's ears or sound receptors. Given the distance between the ears, the difference between the two ToF measurements (interaural time difference, ITD) makes it possible to compute the azimuthal location of the target analytically. Although biological systems are not adapted to solve algebraic equations, they perform localization tasks very efficiently. The barn owl's nervous system makes use of an array of coincidence detector (CD) neurons[35] (i.e., neurons able to detect temporal correlations between spikes propagating down converging excitatory terminals)[38,39] organized into a computational map to solve the localization task.

Previous studies have shown that both complementary metal-oxide-semiconductor (CMOS) and RRAM-based neuromorphic hardware inspired by the inferior colliculus ("auditory cortex") of barn owl constitute an efficient way to compute the position from the ITD[13,40–46]. However, the potential of a full neuromorphic system that couples auditory signals to the neuromorphic computational map has not yet been proven. The main challenge is the intrinsic variability of analog CMOS circuits, affecting the coincidence detection precision. An alternative digital implementation for the ITD estimation has been recently demonstrated[47]. In this work, we propose to exploit the ability of RRAMs to change their conductance value in a nonvolatile manner to counteract the variability in analog circuits. We implemented an experimental system consisting of a single emitting pMUT membrane working at 111.9 kHz, two reception pMUT membranes (sensors) that emulate the barn owl's ears, and a neuromorphic computational map fabricated by co-integrating a 130-nm CMOS processor with hafnium-dioxide RRAM devices. We experimentally characterized the pMUT sensory system and the RRAM-based ITD computational map to validate our localization system and to estimate its angular resolution.

We compared our approach to a microcontroller performing the same localization task using either conventional beamforming or neuromorphic techniques and to the digital implementation on a field-programmable gate array (FPGA) for ITD estimation proposed in ref. [47]. This comparison highlights the competitive energy efficiency of the proposed RRAM-based analog neuromorphic system.

## Results

**Biological background.** One of the most striking examples of precise and efficient object localization systems can be found in barn owls[35,37,48]. At dusk and dawn, barn owls (*Tyto Alba*) actively search for small prey such as voles or mice relying mostly on passive listening. These auditory specialists can locate auditory cues incoming from their prey with astonishing accuracy (about 2°)[35], as shown in Fig. 1a. Barn owls infer the localization of a sound source in the azimuthal (horizontal) plane from the difference between the Time-of-Flight incoming from the source on the two ears (ITD). The ITD computation mechanism has been postulated by Jeffress[49,50], it relies on neural geometry and requires two key ingredients: axons, neuron's nerve fibers, that act as delay lines, and an array of coincidence detector neurons organized into a computational map, as depicted in Fig. 1b. The sound reaches the ears with an azimuth-dependent time delay (ITD). In each ear, the sound is then converted into a spike pattern. Axons from the left and right ears act as delay lines and converge at CD neurons. In theory, only one neuron of the array of coincidence neurons will receive simultaneous inputs (where the delay is compensated exactly), and will fire maximally (neighboring cells will fire too, but at a lower rate). The activation of the particular neuron encodes the position of the target object in space and there is no need to further convert the ITD into an angle. This concept is summarized in Fig. 1c: for example, if the sound originates from the right, a coincidence will occur when the input signal from the right ear travels a longer path than from the left ear by an amount compensating the ITD, e.g. at coincidence neuron 2. In other words, each CD responds to a specific ITD (also called Best Delay) because of axonal delays. In this way, the brain transforms temporal information into spatial information. Anatomical evidence has been found for this mechanism[37,51]. There are phase-locked neurons of the nucleus magnocellularis who preserve the temporal information of the input sound: as their name indicates, they fire at a specific phase of the signal. The coincidence detector neurons of the Jeffress

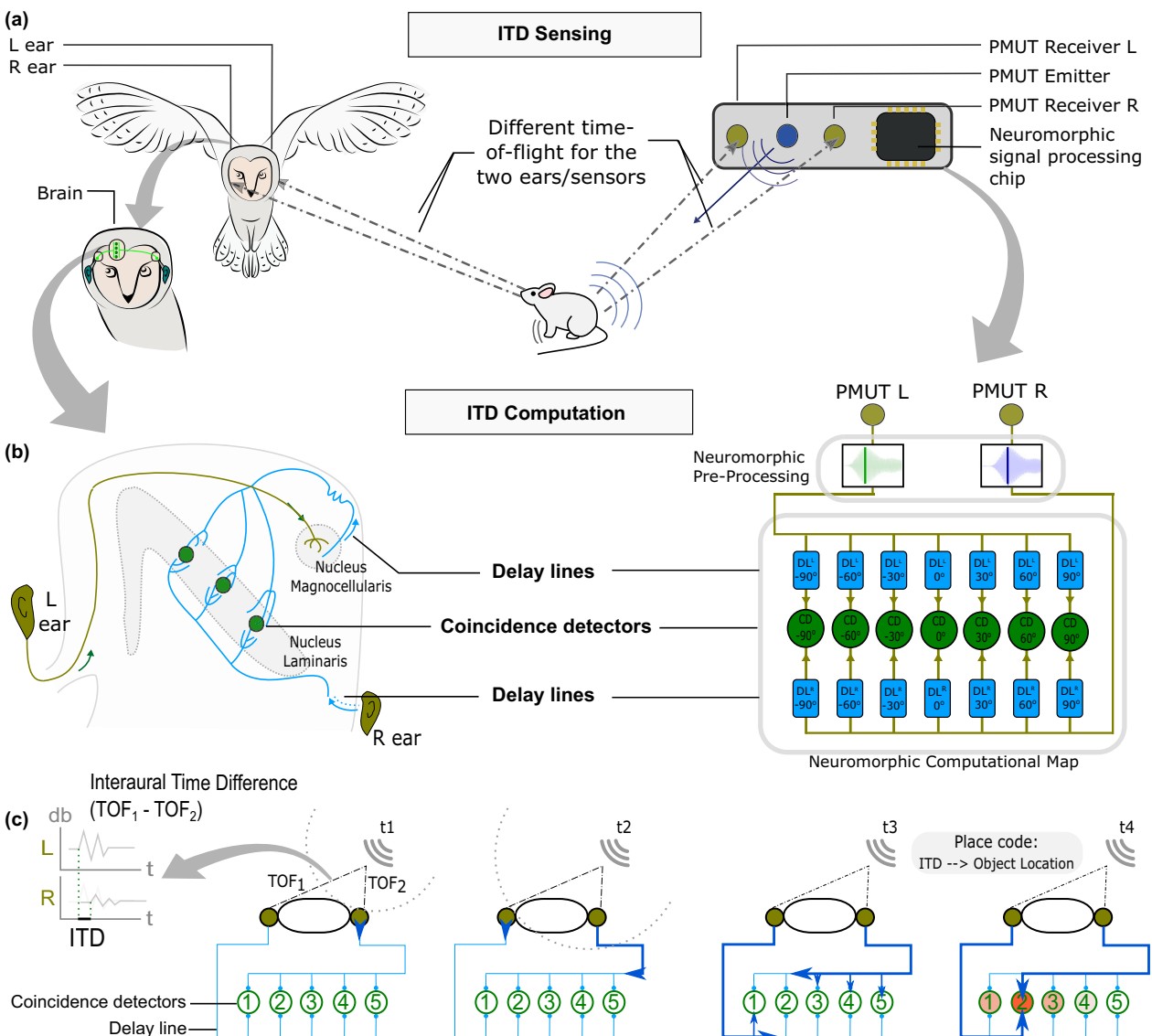

**Fig. 1 Object localization system in barn owls and the proposed bio-inspired technology. a** The barn owl receives a sound wave from a target, a moving prey in this case. The time-of-flight (ToF) of the sound wave at each ear is different (unless the prey is exactly in front of the owl). The dash-dotted lines represent the path of the sound wave toward the barn owl's ears. Based on the difference in the two sound wave path lengths and the corresponding interaural time difference (ITD), it is possible to locate the prey precisely in the horizontal plane (left, figure inspired in ref. [74], Copyright 2002 Society for Neuroscience). In our system, the pMUT emitter (in dark blue) produces a sound wave that bounces off the targeted object. The reflected ultrasound wave is sensed by two pMUT receivers (in light-green) and processed by the neuromorphic processor (right). **b** The ITD computation model (Jeffress model) describes how sounds reaching the barn owl's ears are first encoded into phase-locked spike trains in the nucleus magnocellularis (NM), and then processed using a grid of geometrically arranged coincidence detector neurons in the nucleus laminaris (NL) (left). Illustration of the neural ITD computational map combining delay lines and coincidence detector neurons that can be implemented using RRAM-based neuromorphic circuits to model the owl's biological sensing system (right). **c** Diagram of the basic Jeffress mechanism, where the two ears receive a sound stimulus at different moments due to a difference in the ToF and send axons to detectors from opposite ends. The axons are afferent to an array of coincident detector neurons (CDs), each of which is selective to highly temporally correlated inputs. As a result, only the CDs whose inputs arrive with the smallest time difference (the ITD is exactly compensated) will be maximally excited. The CDs will then encode the angular position of the target.

model can be found in the nucleus laminaris. They receive input from neurons of the nucleus magnocellularis, whose axons serve as delay lines. The amount of delay provided by delay lines may be explained by axonal lengths, but also by differential myelination patterns, changing the conduction speeds. Inspired by the auditory system of the barn owl, we developed a bio-inspired system for object localization. The two ears are represented by the two pMUT receivers. The sound source is a pMUT emitter located in between (Fig. 1a), and the computational map is formed by a grid of RRAM-based CD circuits (Fig. 1b, in green)

taking the role of CD neurons, whose inputs are delayed by delay line circuits (in blue) which act as the axons in the biological counterpart. The proposed sensory system diverges from that of the owl in terms of operation frequency, the barn owl hearing system works in the range 1–8 kHz, but this work uses pMUT sensors that work at around 117 kHz. The choice of an ultrasound sensor was pondered around engineering and optimization criteria. The first one is that restricting the band of reception to—ideally—a single frequency improves the accuracy of the measurement and simplifies the post-processing stage. In addition,

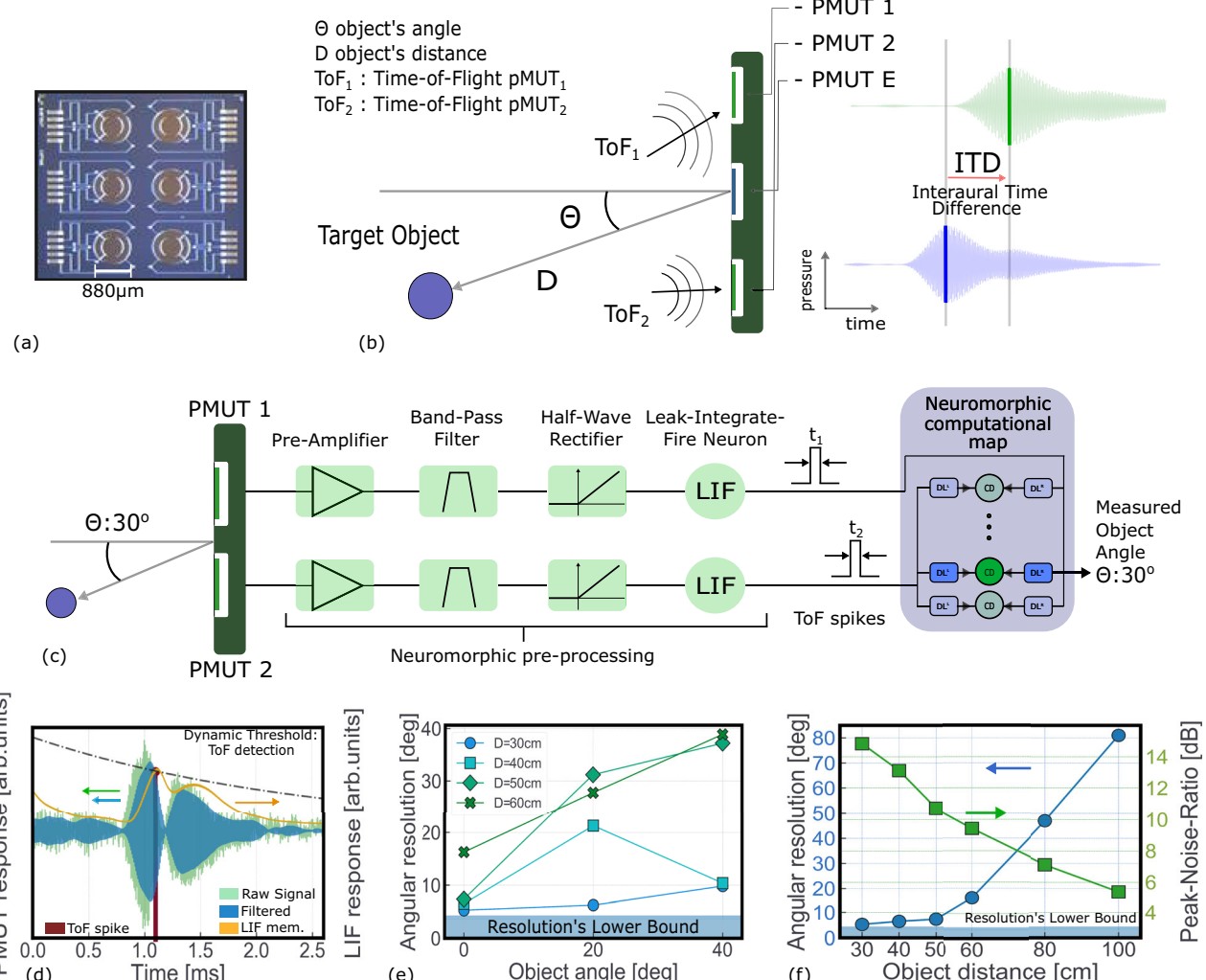

**Fig. 2 Sensory system assessment. a** Picture of a pMUT die with six 880 µm-diameter membranes integrated with a 1.5-mm pitch. **b** Diagram of the measurement setup. A target object is located at an azimuthal position θ and distance D. An emitter pMUT produces a waveform at 117.6 kHz that is reflected by the object and arrives at the two pMUTs receivers with different time-of-flight (ToF). Such difference, defined as interaural time difference (ITD), encodes the position of the object and can be estimated by evaluating the peak of the response in the two receiver sensors. **c** Diagram of the pre-processing steps to convert the raw pMUT signal into a train of spikes (i.e. the input for the neuromorphic computational map). The pMUT sensors and the neuromorphic computational map have been fabricated and tested, while the neuromorphic pre-processing is based on software simulations. **d** Response of the pMUT membrane upon arrival of a signal and conversion to the spike domain. **e** Experimental angular precision of the localization as a function of the object angle (Θ) and the distance (D) of the target object. The minimum angular resolution imposed by the ITD extraction method is of about 4 °C. **f** Angular precision (blue line) and corresponding Peak-to-Noise Ratio (green line) as a function of the object distance for Θ = 0.

operating in the ultrasound regime has the advantage of making the emitted pulse inaudible, and thus not bothersome, for humans as their auditory range is ~20–20 kHz.

**PMUT sensors for time-of-flight measurement**. Piezoelectric micromachined ultrasonic transducers are scalable ultrasound sensors that can be integrated with advanced CMOS technology[31–33,52], and have lower actuating voltage and power consumption than conventional bulk transducers[53]. In our work, the diameter of the membrane is 880 µm and the resonance frequency spreads in the range 110−117 kHz (Fig. 2a, see "Methods" for more details). Over a batch of ten tested devices, the median quality factor is around 50 (ref. [31]). This technology has already reached industrial maturity, and it is not bio-inspired per se. Combining the information of different pMUT membranes is a well-known technique to infer angular information from pMUT devices, using for instance beamforming techniques[31,54].

However, the signal processing required to extract the angular information does not suit low-power sensing. The proposed system, coupling a neuromorphic circuit for pre-processing of the pMUT data and a RRAM-based neuromorphic computational map inspired by the Jeffress model (Fig. 2c), offers an alternative energy-efficient and resource-constrained hardware solution. We conducted an experiment locating two pMUT sensors about 10 cm apart from each other, thus fully taking advantage of the different ToF of sound being sensed by the two receiving membranes. A single pMUT working as an emitter is located in between the receivers. A 12 cm-wide PVC plate located in front of the pMUT devices at a distance D was used as a target (Fig. 2b). The receivers record the sound reflected from the object and respond maximally at the time-of-flight of the sound wave. The experiment was repeated varying the position of the object, defined by its distance D and its angle Θ. Inspired in ref. [55], we propose a neuromorphic pre-processing of the pMUT raw signal to convert the reflected waves into spikes that are the input of the

neuromorphic computational map. The ToF, corresponding to the wave peak amplitude, is extracted from each of the two channels and encoded as the precise timing of a single spike. Figure 2c shows the circuits needed to connect the pMUT sensors to the RRAM-based computational map: for each of the two pMUT receivers, the raw signal is band-pass filtered to smooth it out, rectified, and later passed to a leaky-integrate-and-fire (LIF) neuron that produces an output event (spike) in case of over-coming a dynamical threshold (Fig. 2d): the timing of the output spike encodes the detected time-of-flight. The threshold of the LIF is calibrated to the pMUT response mitigating the pMUT's device-to-device variability. Thanks to this approach, instead of storing the whole sound wave to memory and process it later, we simply generate a spike corresponding to the ToF of the sound wave, which constitutes the input of the resistive memory-based computational map. The spikes are directly sent to the delay lines and coincident detector modules organized in parallel into the neuromorphic computational map. Since they are sent to transistor gates no additional amplification circuitry is required (see Supplementary Fig. 4 for additional details). To assess the localization angular precision allowed by the pMUTs and the proposed signal processing technique, we measured the ITD (i.e., the time difference between the spike events generated by the two receivers) when varying the distance and angle of the object. The ITD is then analytically converted into an angle (see "Methods") and plotted as a function of the object position: the uncertainty over the measured ITDs grows with both the object's distance and angle (Fig. 2e, f). The main challenge is the peak-to-noise ratio (PNR) in the pMUT response. The more distant the object is located, the lower the acoustic signal, thus lowering the PNR (Fig. 2f, green line). The decrease in the PNR leads to an increase in the uncertainty over the estimated ITD and consequently on the precision of the localization (Fig. 2f, blue line). For an object located 50 cm away from the emitter, the system's angular precision is about 10°. This limit, imposed by the sensor's characteristics, can be improved. For example, the pressure sent by the emitter can be increased raising up the voltage driving the pMUT's membrane[34]. Another solution to strengthen the emitted signal is to couple several emitters[56]. These solutions would extend the range of detection at the price of an added energy cost. Other improvements can be performed on the receiving side. The noise floor of the pMUT receiver could be drastically reduced improving the connections between the pMUT and the first stage amplifier, currently done using wire-bonding and RJ45 cables.

**RRAM-based neuromorphic computational map.** Resistive memories store information in their nonvolatile conductive states. The basic working principle of this technology is that modifying a material at the atomic level results in changes of its conductance[57]. Here, we use an oxide-based resistive memory composed of a 5 nm hafnium-dioxide layer sandwiched between a top and a bottom electrode of titanium and titanium nitride. The conductivity of an RRAM device can be modified by the application of current/voltage waveforms, which create or break a conductive filament composed of Oxygen vacancies between the electrodes. We have co-integrated such devices in a standard 130 nm CMOS process[58] to build a fabricated re-configurable neuromorphic circuit implementing coincidence detectors and the delay lines circuits (Fig. 3a). Both the non-volatility and analog nature of the devices perfectly couple with the event-driven nature of the neuromorphic circuits, minimizing power consumption. The circuit has an instant on/off feature: it works immediately after being turned on, allowing to cut the power supply entirely as soon as the circuit is idle. The basic building block of the proposed circuit is presented in Fig. 3b. It is

composed of $N$ parallel one-resistor-one-transistor (1T1R) structures, encoding the synaptic weights, from which a weighted current is extracted and then injected to a common differential pair integrator (DPI) synapse[59], and finally into a leaky-integrate-and-fire (LIF) neuron[60] (see "Methods" for more details). The input spikes are applied at the gates of the 1T1R structures as trains of voltage pulses, with a pulse width on the order of hundreds of nanoseconds. The resistive memories can be SET into a high conductance state (HCS) by applying an external positive voltage reference on $V_{top}$ while grounding $V_{bottom}$, and RESET into a low conductive state (LCS) by applying a positive voltage on $V_{bottom}$ while grounding $V_{top}$. The mean value of the HCS can be controlled by limiting the SET programming (compliance) current ($I_{CC}$) via the gate-source voltage of the series transistor (Fig. 3c). The function of RRAMs in the circuit is dual: they route and weigh input pulses.

First, thanks to the two main conductive states (HCS and LCS), the RRAMs can either block or pass the input pulses when they are respectively in the LCS or HCS state. As a result, RRAMs efficiently define the connections in the circuit. This is fundamental to allow the architecture to be re-configurable. In order to prove that, we characterized a fabricated circuit implementation of the circuit block in Fig. 3b. An RRAM corresponding to $G_0$ was programmed into the HCS and a second RRAM, $G_1$, was programmed in the LCS. Input pulses were applied to both $V_{in0}$ and $V_{in1}$. The effect of two input trains of pulses was analyzed in the output neuron, by collecting the membrane voltage and output of the neuron with an oscilloscope. The experiment is successful when only the pulses connected to the neuron by the HCS device ($G_0$) excite the membrane voltage. This is demonstrated in Fig. 3d, where the blue train of pulses makes the membrane voltage accumulate charge on the membrane capacitor, whereas the green train of spikes leaves the membrane voltage unperturbed.

The second important function of RRAMs is implementing the weight of the connections. By exploiting the analog adjustment of the RRAMs conductance, the input-to-output connection can be appropriately weighted. In a second experiment, the device $G_0$ is programmed in different HCS levels and an input pulse is applied to the input $V_{In0}$. The input pulse extracts a current from the device ($I_{weight}$) which is proportional to the conductance and the corresponding potential drop $V_{top} - V_{bot}$. This weighted current is then injected into the DPI synapse and output LIF neuron. The membrane voltage of the output neuron is recorded with an oscilloscope and plotted in Fig. 3e. The peak of the neuron membrane voltage responding to a single input pulse is proportional to the conductance of the resistive memory, confirming that RRAMs can be exploited as programmable synaptic weight elements. These two preliminary tests demonstrate that the proposed RRAM-based neuromorphic platform is able to implement the basic elements of the Jeffress basic mechanism, namely the delay line and coincidence detector circuits. The circuital platform is constituted by stacking consecutive blocks, as the one in Fig. 3b, side-by-side and connecting their gates to common input lines. We designed, fabricated, and tested a neuromorphic platform composed of two output neurons and receiving two inputs (Fig. 4a). The layout of the circuit is shown in Fig. 4b. The upper $2 \times 2$ RRAM matrix allows to route of the input pulses to the two output neurons, while the lower $2 \times 2$ array allows the two neurons ($N_0$, $N_1$) to be recurrently connected. We demonstrate that this platform can assume a delay line configuration and two distinct coincidence detector functionalities, as summarized by the experimental measurements in Fig. 4c–e.

The delay line (Fig. 4c) simply exploits the dynamical behavior of the DPI synapse and LIF neuron to reproduce the input spike

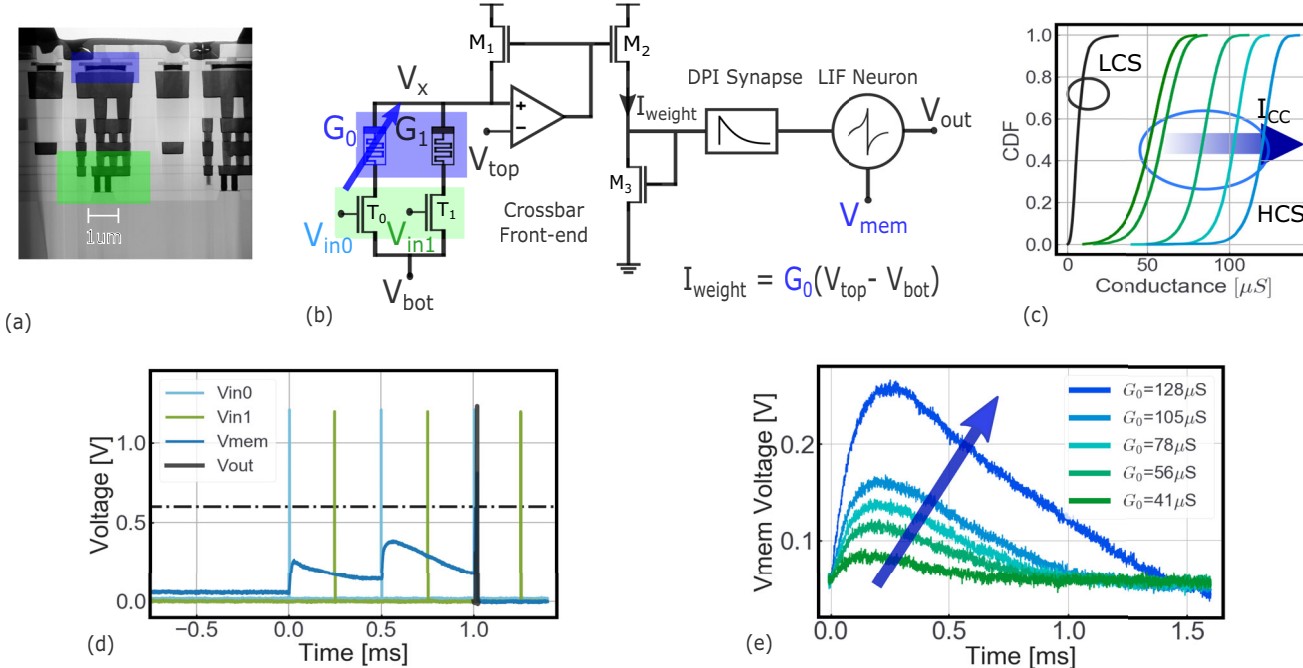

**Fig. 3 Role of RRAM devices in neuromorphic circuits. a** Scanning Electron Microscopy (SEM) image of a HfO$_2$ 1T1R RRAM device, in blue, integrated on 130 nm CMOS technology, with its selector transistor (width of 650 nm) in green. **b** Basic building block of the proposed neuromorphic circuit. Inputs voltage pulses (spikes), $V_{in0}$ and $V_{in1}$, draw a current $I_{weight}$ proportional to the conductance states, $G_0$ and $G_1$, of the 1T1R structures. This current is injected into a DPI synapse that excites a LIF neuron. The RRAMs $G_0$ and $G_1$ are set in the HCS and LCS, respectively. **c** Cumulative density function of the conductance of a population of 16 kb RRAM devices, as a function of the compliance current $I_{CC}$, which effectively controls the conductance level. **d** Measurement of the circuit in (**a**), showing that $G_1$ (in LCS) effectively blocks inputs from $V_{in1}$ (green), in fact the output neuron's membrane voltage only responds to the blue input of $V_{in0}$. RRAMs efficiently define the connections in the circuit. **e** Measurement of the circuit in (**b**) showing the effect of the conductance value $G_0$ on the membrane voltage $V_{mem}$, following the application of a voltage pulse $V_{in0}$. The larger the conductance, the stronger the response: the RRAM device thus implements the weight of the input-to-output connection. Measurements have been performed on one circuit and demonstrate the dual function of the RRAMs, they route and weigh the input pulses.

from $V_{in1}$ to $V_{out1}$ with a delay $T_{del}$. Only the RRAM connecting $V_{in1}$ to $V_{out1}$, $G_3$, is programmed into the HCS, while the other RRAMs are in the LCS. The $G_3$ device is programmed to 92.6 μS to ensure that each input spike increases the membrane voltage of the output neuron sufficiently to reach the threshold and to generate a delayed output spike. The delay $T_{del}$ is defined by both the synapse and neuron time constants. A coincidence detector detects the occurrence of temporally correlated but spatially distributed input signals. A direction insensitive CD relies on separate inputs converging to a common output neuron (Fig. 4d). The two RRAMs connecting $V_{in0}$ and $V_{in1}$ to $V_{out1}$, $G_2$ and $G_4$, respectively, are programmed into the high conductance state. The synchronous arrival of spikes at $V_{in0}$ and $V_{in1}$ pushes the membrane voltage of the neuron $N_1$ over the threshold required to generate an output spike. If the two inputs arrive too far apart in time, the charge on the membrane voltage accumulated by the first input may have time to decay away, preventing the membrane potential of $N_1$ to reach the threshold. The $G_1$ and $G_2$ are programmed to around 65 μS, ensuring that a single input spike does not increase the membrane voltage enough to generate an output spike. Coincidence detection between spatially and temporally distributed events is a basic operation common to a wide range of sensing tasks, such as optical flow-based obstacle avoidance[61], and sound source localization. Consequently, the computation of both direction-sensitive and -insensitive CDs are fundamental building blocks to build for both visual and sound localization systems. The proposed implementation of the circuit fits a range of four orders of magnitude of time scales, as shown by the characterization of the time constant (see Supplementary Fig. 2). It can therefore fit with the requirements of both visual

and sound systems. The direction-sensitive CD is a circuit sensitive to the spatial order of arrival of impulses: from right to left or vice versa. This is a basic building block in the elementary motion detection network of Drosophila's visual system to compute the direction of motion and detection of collisions[62]. To implement a direction-sensitive CD the two inputs have to be routed to two different neurons ($N_0$, $N_1$) and between those, a directional connection has to be established (Fig. 4e). Upon the arrival of the first input, $N_0$ responds by increasing its membrane voltage up to overcoming its threshold and emitting a spike. Thanks to the directional connection in green, this output event in turn excites $N_1$. If the $V_{in1}$ input event arrives to excite $N_1$ when its membrane voltage is still high, $N_1$ will produce an output event, signifying the detection of coincidence between the two inputs. The directional connection allows $N_1$ to emit an output only if input 1 arrives after input 0. The $G_0$, $G_3$, and $G_7$ are, respectively, programmed to 73.5 μS, 67.3 μS, and 40.2 μS, ensuring that a single input spike at $V_{in0}$ generates a delayed output spike, while the membrane potential of $N_1$ reaches the threshold only upon the synchronous arrival of two input spikes.

**Variability in neuromorphic circuits and RRAM-based calibration procedure.** Variability is a source of non-ideality in analog neuromorphic systems[63–65]. It results in heterogeneous behaviors among neurons and synapses. Examples of such imperfections include for example 30% (mean value over standard deviation) of variability on input gain, time constants and refractory period, to name a few (see "Methods"). This issue is more pronounced when several neuron circuits are connected

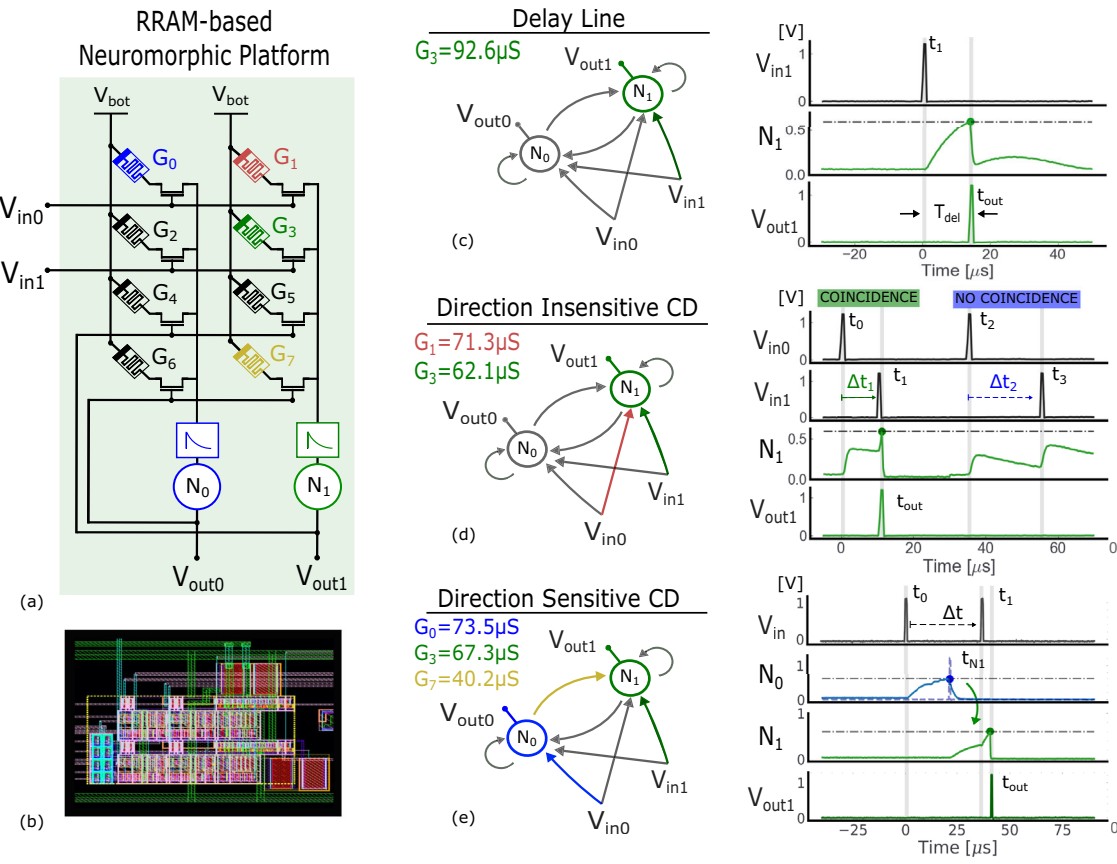

**Fig. 4 Experimental measurements of the RRAM-based neuromorphic circuital platform. a** Diagram of the circuit formed by two output neurons, $N_0$ and $N_1$, receiving two inputs 0 and 1. The four devices on the top of the array define the synaptic connections from inputs to outputs and the four cells on the bottom define the recurrent connections among neurons. Colored RRAMs represent devices set in the HCS on the diagrams to the right: a device in the HCS allows a connection to be formed and expresses a weight, while a device in the LCS blocks the input pulse and disables the connection to the output. **b** Layout of the circuit in (**a**), with the eight RRAMs highlighted in blue. **c** A delay line is formed by simply exploiting the dynamics of a DPI Synapse and a LIF Neuron. The green RRAM is set to high enough conductance to allow the output spike to be elicited following the input spike by a delay $\Delta t$. **d** Diagram of the direction insensitive CD detecting temporally correlated signals. The output neuron 1, $N_1$, spikes upon arrival of input 0 and input 1 with small delay. **e** Diagram of the direction-sensitive CD, a circuit that detects when input 1 arrives in close proximity and after input 0. The output of the circuit is represented by neuron 1 ($N_1$).

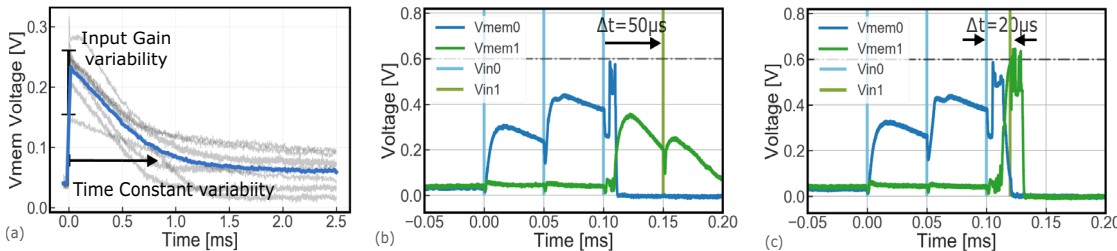

**Fig. 5 Variability in analog neuromorphic circuits. a** Experimental measurements of the response of nine randomly selected individual neurons to the same input spike. The response varies across the population impacting both the input gain and the time constant. **b** Experimental measurements of the impact of the neuron to neuron variability affecting the direction-sensitive CD. Due to the neuron-to-neuron variability, the two output neurons of the direction-sensitive CD respond differently to the input stimuli. Neuron 0 presents a lower input gain than neuron 1, thus requiring three input pulses (instead of 1) to produce an output spike. Neuron 1 reaches the threshold with two input events, as expected. If input 1 arrives $\Delta t = 50\,\mu s$ after neuron 0 has been excited, the CD remains silent because $\Delta t$ is larger than neuron 1's time constant (about 22 $\mu s$). **c** Decreasing the $\Delta t = 20\,\mu s$, makes input 1's spike to arrive when neuron 1's excitation is still high, resulting in the coincidence detection of the two input events.

together, as in the case of the direction-sensitive CD, which consists of two neurons. To function properly, the input gain and decay time constants of the two neurons should be as similar as possible. For example, large differences in input gain may result in a neuron responding excessively to an input pulse, while the other being almost insensitive. Figure 5a shows that randomly

selected neurons respond differently to the same input pulse. This neuron variability has an impact on e.g. the functionality of the direction-sensitive CD. In the circuit characterized in Fig. 5b, c, neuron 1 presents a much higher input gain than neuron 0. As a result, neuron 0 requires three input pulses (instead of 1) to reach the threshold, while neuron 1 reaches the threshold with two

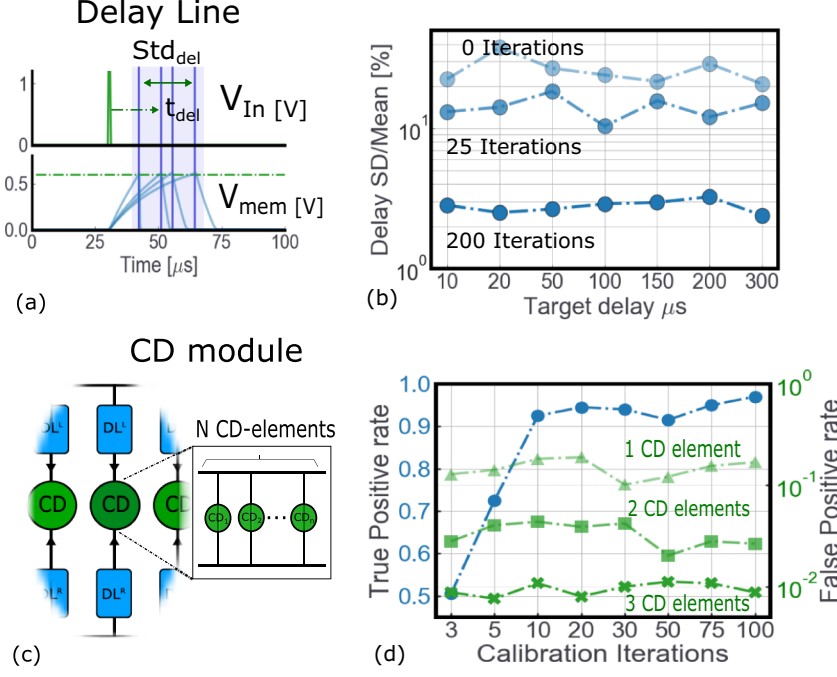

**Fig. 6 Performance of the delay line and direction insensitive CD circuits and impact of the RRAM calibration procedure. a** Impact of the neuron variability on the delay line circuit. **b** The delay line circuit can be scaled up to larger delays, setting the time constant of the corresponding LIF neuron and DPI synapse to larger values. Increasing the iterations of the RRAM calibration procedure enabled us to substantially improve the precision on the target delay: 200 iterations allow the error to be reduced to less than 5%. One iteration corresponds to a SET/RESET operation on the RRAM cell. **c** Each CD module in the Jeffress model can be implemented with $N$-parallel CD elements, to be more resilient to system failures. **d** More RRAM calibration iterations allow to improve the true positive rate (blue line), while the false-positive rate is independent of the number of iterations (green lines). Stacking more CD elements in parallel enabled us to avoid false coincidence detection from a CD Module.

input events, as expected. The implementation of the bio-inspired spike-timing-dependent plasticity (STDP) is a possible way to mitigate the impact of the imprecise and sluggish neuron and synapse circuits on the system performances[43]. Here, we propose to exploit the plastic behavior of the resistive memory as a mean of acting on the input gain of neurons and reduce the impact of the neuromorphic circuit variability. As demonstrated in Fig. 4e, the conductance level associated with an RRAM synaptic weight effectively modulates the response of the associated neuron membrane voltage. We adopt an iterative RRAM programming strategy. For a given input, the conductance value of the synaptic weights is re-programmed until the target circuit behavior is obtained (see "Methods").

The two elements employed in the ITD computational map are the delay lines and direction insensitive CD. Both circuits need to be precisely calibrated to ensure the good performance of the object localization system. The delay line has to precisely deliver a delayed version of the input spike (Fig. 6a), the CD must be activated only when the inputs fall within the target detection range. For the delay line, the synaptic weight of the input connection ($G_3$ in Fig. 4a) is re-programmed until the target delay is obtained. A tolerance around the target delay is set to stop the procedure: the smaller the tolerance, the harder it is to successfully tune the delay line. Figure 6b shows the result of the calibration procedure for the delay line: as it can be seen the proposed circuit can precisely provide all the delays required in the computational map (from 10 to 300 µS). The maximum number of calibration iterations affects the quality of the calibration procedure: 200 iterations allow the error to be reduced to less than 5%. One calibration iteration corresponds to a SET/RESET operation of the RRAM cell. The tuning procedure is also crucial to improve the accuracy of the detection of the temporally close events of the CD module. Ten calibration iterations are

needed to reach a true positive rate (i.e., rate of events correctly detected as correlated) higher than 95% (blue line in Fig. 6c). However, the tuning procedure has no effect on false-positive events (i.e., rate of events incorrectly detected as correlated). Another technique observed in biological systems which solves the time constraint of rapidly activated pathways is redundancy (i.e., many copies of the same entity are used to fulfill a given function). Taking inspiration from biology[66], we stacked multiple CD circuits in each CD module between two delay lines to reduce the effect of false-positive detection. As shown in Fig. 6c (green lines), stacking three CD elements in each CD module allows reducing the false-positive rate to less than $10^{-2}$.

**System assessment**. We now assess the performance and power consumption of the end-to-end integrated object localization system presented in Fig. 2 using the measured results of the acoustical characterization of the pMUT sensors, the CD, and the delay line circuits composing the neuromorphic computational map inspired by the Jeffress model (Fig. 1a). Regarding the neuromorphic computational map, the higher the number of CD modules, the better the angular resolution, but also the higher the system energy (Fig. 7a). A trade-off is reached by comparing the precision of the single components (both pMUT sensors, and neuron and synapse circuits) with that of the whole system. The resolution of delay lines are limited by the time constants of the analog synapses and neurons, which are greater than 10 µs in our circuits, corresponding to an angular resolution of 4° (see "Methods"). A more advanced CMOS technology node would enable the design of neuron and synapse circuits with lower time constants and consequently higher precision of the delay line element. However, in our system, the precision is limited by the pMUT uncertainty in the estimation of the angular position, that

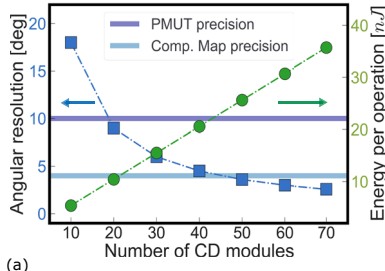
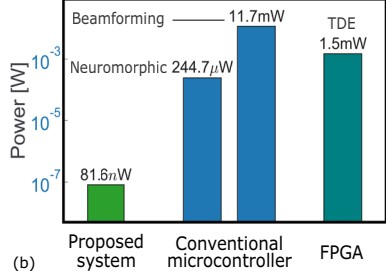

**Fig. 7 Power consumption and angular resolution of the presented neuromorphic sensory and signal processing system. a** Angular resolution (blue) and energy consumption (green) of one localization operation as a function of the number of CD modules. The dark-blue horizontal bar represents the angular precision of the PMUTs while the light-blue horizontal bar represents the angular precision of the neuromorphic computational map. **b** Power consumption of the proposed system and comparison with the two discussed implementations on a microcontroller and the digital implementation on an FPGA of the temporal-difference encoder (TDE)[47].

is 10º (dark-blue horizontal line in Fig. 7a). We fixed the number of CD modules to 40, corresponding to an angular resolution of about 4º, that is the computational map angular precision (light-blue horizontal line in Fig. 7a). At the system level, this results in a 4º resolution and 10º precision for an object located in front of the sensory system at a distance of 50 cm. This value is comparable to the sound localization neuromorphic system reported in ref. [67]. A comparison of the proposed system with the state-of-the-art can be found in Supplementary Table 1. Adding extra pMUTs, boosting the acoustic signal level, and bringing down the electronic noise are possible ways to further improve the localization precision. The single bank power consumption for the neuromorphic pre-processing of the pMUT signal (see Fig. 2) is evaluated at 9.7 nW, according to ref. [55]. Accounting for the 40 CD modules in the computational map, the energy per operation (i.e. energy to localize an object) estimated by SPICE simulations is 21.6 nJ. The neuromorphic system is activated only at the arrival of an input event, i.e., when the sound wave reaches any of the pMUT receivers and overcomes the detection threshold, and kept idle otherwise. This allows avoiding unnecessary energy consumption when no input signal is present. Considering a rate of localization operations of 100 Hz and an activation period of 300 µs per operation (maximum possible ITD), the power consumption of the neuromorphic computational map results being of 61.7 nW. Accounting for the neuromorphic pre-processing applied to each of the pMUT receivers brings the total system's power consumption to 81.6 nW. To gain a perspective on the energy efficiency of the proposed neuromorphic approach compared to conventional hardware, we benchmark this figure to the energy required for running the same task on a state-of-the-art low-power microcontroller[68] using either neuromorphic or conventional beamforming techniques. The neuromorphic method accounts for an analog-digital-converter (ADC) stage followed by a Band-Pass filter and an envelope extraction stage (Teager–Kaiser method). Finally, a thresholding operation is performed to extract the ToF. We omit the computation of the ITD based on the ToF and the conversion to the estimated angular position as it happens once per measurement (see "Methods"). Assuming a sampling frequency of 250 kHz on the two channels (pMUT receivers), 18 operations for the band-pass filter, 3 operations for the envelope extraction, and 1 operation for the thresholding per sample, the estimate of the overall power consumption leads to 245 µW. This leverages the microcontroller's low-power mode[69] which is enabled when not performing the algorithm, reducing the power consumption to 10.8 µW. The power consumption for the beamforming signal processing solution proposed in ref. [31], accounting for 5 pMUT receivers and 11 beams equally distributed in the [−50º, +50º] azimuthal plane is 11.71 mW (see "Methods" for the details).

Furthermore, we report the power consumption of the FPGA-based Temporal-Difference-Encoder (TDE)[47], as an alternative to the Jeffress model for object localization, evaluated at 1.5 mW. Based on these estimations, the proposed neuromorphic approach achieves a reduction of five orders of magnitudes in power consumption relative to a microcontroller adopting a classical beamforming technique for an object localization operation. Adopting a neuromorphic signal processing approach on a classical microcontroller reduces the power consumption of about two orders of magnitude. The efficiency of the proposed system can be attributed to the combination of the asynchronous, resistive memory-based analog circuits able to perform in-memory computing, along with the absence of analog-to-digital conversion of the sensed signal.

## Discussion

To minimize the energy consumption of the object localization system, we envisioned, designed, and implemented an efficient, event-driven RRAM-based neuromorphic circuit that processes signal information produced by embedded sensors to calculate a targeted object's position in real time. Whereas conventional processing techniques would continuously sample the detected signal and crunch calculations to extract the useful information, the proposed neuromorphic solution computes asynchronously as the useful information arrives: this has allowed us to increase the system's energy efficiency by five orders of magnitude. Furthermore, we highlight the flexibility of RRAM-based neuromorphic circuits. The capability of RRAMs to change their conductance in a nonvolatile manner (plasticity) compensates for the inherent variability of the ultra-low-power analog DPI synapse and neuron circuits. This makes such RRAM-based circuits versatile and powerful. Our goal is not to extract complex features or patterns from a signal, but to locate an object in real time. Our system is also capable of efficiently compressing the signal and eventually sending it to a subsequent processing stage, if needed, for more complex decision-making. In the context of a localization application, our neuromorphic pre-processing stage can provide information about the position of a target. This information can be used, for example, for motion detection or gesture recognition. We stress the importance of coupling ultra-low-power sensors, such as pMUTs, with ultra-low-power electronics. To do so, the neuromorphic approach has been key, as it led us to the design of novel circuit implementations of biologically-inspired computational methods such as the Jeffress model. In the context of sensor fusion application, our system can be coupled with several different event-based sensors to compute more accurate information. Although owls can perfectly locate their prey in the dark, they have an excellent vision, and prey

capture is preceded by combined auditory and visual search[70]. When a specific auditory neuron fires, the owl has the information needed to determine in which direction to start the visual search, focusing its attention on only small subsets of the visual scene. Combining visual sensors (DVS cameras) and the proposed audition sensor (pMUT based) should be explored to develop future autonomous agents.

## Methods

**Acoustic measurement setup and pMUT characterization.** pMUT sensors are arranged in a printed circuit board, separating the two receivers of about 10 cm, with the emitter between the receivers. In this work, each membrane is a suspended bimorph structure made of two 800-nm-thick piezoelectric aluminum nitride (AlN) layers sandwiched between three 200-nm-thick Molybdenum (Mo) layers, covered by a 200-nm-thick top SiN passivation layer, as reported in ref. [71]. Inner and outer electrodes are patterned on the bottom and the top Mo layers, while the middle Mo electrode is not patterned and used as the ground, resulting in a membrane with four electrode pairs.

This architecture enables to exploit the whole deformation of the membrane, resulting in a enhanced drive and receive sensitivity. Such a pMUT typically presents a drive sensitivity of typically 700 nm/V as an emitter, delivering a surface pressure of 270 Pa/V. As a receiver, a single pMUT membrane presents a short-circuit sensitivity of 15 nA/Pa, directly related to the piezoelectric coefficients of AlN. The technological variability of the stress within the AlN layers results in a resonant frequency variation which is compensated by applying a DC bias to the pMUT. The DC sensitivity has been measured at 0.5 kHz/V. For acoustic characterization, a microphone is used in front of the pMUT.

For pulse-echo measurements, we positioned a rectangular plate of about 50 cm$^2$ in front of the pMUTs, reflecting the emitted sound wave. Both the distance of the plate and the angle with respect to the pMUT plane are controlled utilizing dedicated supports. Tectronix CPX400DP voltage sources bias the three pMUT membranes to tune the resonant frequency to 111.9 kHz[31], while the emitter is controlled by a Tectronix AFG 3102 pulse generator set close to the resonance frequency (111.9 kHz), and a duty cycle of 0.01. The currents read at the four output ports of each pMUT receiver are converted into a voltage by a dedicated differential current-to-voltage architecture and the resulting signal is digitized by a Spektrum acquisition system. We characterized the limit of detection by collecting the pMUT signal in different conditions: we moved the reflecting plate at different distances [30, 40, 50, 60, 80, 100] cm and varied the angle of the pMUT support ([0, 20, 40]°). Figure 2b shows the relationship between the temporal resolution in detecting ITD and the corresponding angular position in degrees.

**Design and fabrication of neuromorphic circuits.** Two different fabricated RRAM circuits are used in this article. The first is a 16,384 (16k) device array (128 × 128 devices) of one-transistor/one-resistor, 1T1R, structures. The second chip is the neuromorphic platform presented in Fig. 4a. The RRAM cell consists of HfO$_2$ 5-nm-thick film sandwiched in a TiN/HfO$_2$/Ti/TiN stack. The RRAM stack is integrated into the back end Of Line (BEOL) of a standard 130 nm CMOS process. RRAM-based neuromorphic circuits present the challenge of designing a fully analog electronic system in which the RRAM devices coexist with conventional CMOS technology. In particular, the conductive state of the RRAM devices has to be read and utilized as a functional variable of the system. To do so, a circuit that reads a current from a device upon arrival of an input pulse and that uses such current to weight the response of a differential pair integrator (DPI) synapse has been designed, fabricated, and tested. The circuit is shown in Fig. 3a and it represents the basic building block of the neuromorphic platform in Fig. 4a. The input pulses activate the gate of the 1T1R devices, resulting in a current flow through the RRAM proportional to the conductance of the device, G ($I_{weight} = G(V_{top} − V_x)$). The operational amplifier circuit (OPAMP) has a constant DC bias voltage $V_{Top}$ applied to its inverting input. The negative feedback of the OPAMP will act to ensure that $V_x = V_{top}$, by sourcing an equal current from transistor $M_1$. The current extracted from the device, $I_{weight}$, is injected onto the DPI synapse. Stronger currents will result in greater depolarization, thus the RRAM's conductance effectively implements the synaptic weight. This exponential synaptic current is injected onto the membrane capacitor of a leaky-integrate and fire (LIF) neuron where it integrates as a voltage. If the threshold voltage of the membrane (the switching voltage of an inverter) is overcome, the output section of the neuron is activated, producing an output spike. This pulse feeds back and shunts the neuron membrane capacitor to the ground such that it is discharged. The circuit is then complemented by a pulse extender, not shown in Fig. 3a, that reshapes the output pulse of the LIF neuron to the target pulse width. Further multiplexers were integrated on each line in order to be able to apply voltages to the top and bottom electrodes of the RRAM devices.

**Circuit measurement setup and RRAM characterization.** The electrical tests involved analyzing and recording the dynamical behavior of analog circuits as well as programming and reading RRAM devices. Both phases required dedicated instrumentation, all simultaneously connected to the probe card. RRAMs devices in

the neuromorphic circuits are accessed from the external instrumentation by means of multiplexars (MUXs). The MUXs decouple the 1T1R cell from the rest of the circuit where they belong, allowing to read and/or program the device. For programming and reading the RRAM devices, a Keithley 4200 SCS machine was used combined with an Arduino microcontroller: the first for precise pulse generation and current reading, the second to fast access a single 1T1R element in the memory array. The first operation is the forming of the RRAM devices. The cells are selected one by one and a positive voltage was applied between the top and bottom electrodes. At the same time, the current is limited to the order of tens of micro-amperes by applying an appropriate gate voltage to the selector transistor. Afterward, the RRAM cells can be cycled between the low conductance state (LCS) and the high conductance state (HCS) through RESET and SET operations, respectively. SET operations are performed with a positive square voltage pulse of 1 μs width and 2.0−2.5 V peak voltage applied to the Top Electrode, and a similarly shaped synchronous pulse with 0.9−1.3 V peak voltage applied to the gate of the selector transistor. Such values allow to modulate the RRAM conductance in the 20−150 μS interval. For the RESET, a pulse of 1 μs width and 3 V peak is applied to the bottom electrode (Bit Line) of the cell while the gate voltage is in the 2.5−3.0 V range. Inputs and outputs of the analog circuits are dynamical signals. In the case of the input, we have alternated two HP 8110 pulse generators with a Tektronix AFG3011 waveform generator. Input pulses have a width of 1 μs and rise/fall edge of 50 ns. This type of pulse is assumed as the stereotypical spiking event in the spike-based analog circuit. Concerning the outputs, a 1 GHz Teledyne LeCroy oscilloscope was utilized to record the output signals. The acquisition speed of the oscilloscope has been proven not to be a limiting factor analyzing and collecting data from the circuits.

**Variability in neuron and synapse circuits.** Exploiting analog electronics' dynamics to mimic the behavior of neurons and synapses is an elegant and efficient solution to improve computation efficiency. The downside of such a computational substrate is that it is subject to variability, from circuit to circuit. We quantified the variability in neuron and synapse circuits (Supplementary Fig. 2a, b). Of all the manifestations of variability, the most impactful at the system level are the ones concerning the time constant and the input gain. The time constant of LIF neurons and DPI synapses is defined by an $RC$ circuit, where the $R$ value is controlled by a bias voltage applied to a transistor's gate ($V_{lk}$ for neurons, $V_{tau}$ for synapses), defining the leakage rate. Input gain is defined as the peak voltage reached by the synapse and neuron membrane capacitor stimulated by an input pulse. The input gain is controlled by another biased transistor modulating the input current. Monte Carlo Simulations calibrated on ST Microelectronics 130 nm process are performed in order to collect some statistics about the input gain and the time constant. The results are plotted in Supplementary Fig. 2 where input gain and time constant are quantified as a function of the bias voltage controlling the leakage rate. Green markers quantify the standard deviation of the time constants with respect to the mean value. Both neuron and synapse circuits are capable of expressing a wide range of time constants, in the $10^{-5}−10^{-2}$ s range, as shown in Supplementary Fig. 2c, d, while the variability is quantified at 30% for both the neuron and synapse circuit. The input gain (Supplementary Fig. 2e, d) variability is around 8% and 3% for the neuron and synapse respectively. Such defects are well documented in the literature: in the family of DYNAP chips, different measurements have been performed to evaluate the mismatch across a population of LIF Neurons[63]. Synapses in the Mixed-Signal chip BrainScale have been measured and the mismatch between them analyzed, also proposing a calibration procedure to reduce the variability impact at the system level[64].

**RRAM-based calibration procedure.** The function of RRAMs in the neuromorphic circuit is dual: defining the architecture (routing inputs to outputs) and implementing synaptic weights. The latter property can be exploited to mitigate the problem of variability in analog neuromorphic circuits. We developed a simple calibration procedure that consists in re-programming the RRAM device until the circuit under analysis meets certain requirements. For a given input, the output is monitored and the RRAMs are re-programmed until the target behavior is achieved. A 5 s waiting time is introduced between programming operations to mitigate the RRAM Relaxation issue, causing temporal fluctuations of the conductance (Supplementary Information). The synaptic weights are adapted or calibrated to the requirements of the analog neuromorphic circuit. Focusing on the two basic functionalities of the Neuromorphic Platform, the delay lines, and direction insensitive CD, the calibration procedure is summarized in Supplementary Algorithms [1, 2]. For the delay line circuit, the target behavior is to deliver the output pulse with a delay Δt. If the actual delay of the circuit is smaller than the target, the synaptic weight $G_3$ has to be decreased (the $G_3$ has to be RESET and then SET with a lower compliance current, $I_{cc}$). Contrarily, if the actual delay is longer than the target, the conductance of $G_3$ has to be reinforced (the $G_3$ has to be RESET and then SET with a higher $I_{cc}$). This procedure is repeated until the delay produced by the circuit matches the target, with a tolerance set to stop the calibration procedure. For the direction insensitive CD, the calibration procedure involves two RRAM devices, $G_1$ and $G_3$. The circuit is provided with two inputs, $V_{in0}$ and $V_{in1}$ delayed by $dt$. The circuit must only respond to delays lower than the coincidence range $[0,dt_{CD}]$. When an output spike is absent, whereas the input spikes are close, both the RRAM devices must be reinforced in order to help the

neuron reach the threshold. Conversely, if the circuit responds to delays larger than the target range $dt_{CD}$, the conductances have to be decreased. The procedure is repeated until the correct behavior is obtained. The compliance current can be modulated by the embedded analog circuit presented in refs. [72,73]. Exploiting this embedded circuit, one could perform such procedure periodically to calibrate the system or to re-purpose it for different applications.

**Estimation of the power consumption on a microcontroller**. We estimate the power consumption of a neuromorphic signal processing approach on an off-the-shelf 32-bits microcontroller[68]. In this estimation, we assumed to operate with an identical setup to the one presented in this work, with one pMUT emitter and two pMUT receivers. The method accounts for a band-pass filter followed by an envelope extraction stage (Teager–Kaiser method), and finally a thresholding operation applied to the signal to extract the time-of-flight. The computation of the ITD and its conversion to the detected angle are omitted in the estimation. We consider the band-pass filter to be implemented with a 4th order Infinite Impulse Response filter, requiring 18 floating-point operations. Envelope extraction makes use of further three floating-point operations, and a final operation is due for thresholding. In total, 22 floating-point operations are required for pre-processing the signal. The signal sent is a short pulse of a 111.9 kHz sine wave, produced every 10 ms, resulting in a 100 Hz localization operation frequency. We adopt a 250 kHz sample rate to respect the Nyquist theorem and a 6-ms window per measurement to capture a range of 1 m. Note that 6 ms is the time-of-flight for an object located at 1 m distance. This gives a power consumption of 180 μW for the analog-to-digital conversion of 0.5 MSPS. The pre-processing of the signal accounts for 6.60 MIPS (instructions per second), resulting in 0.75 mW. However, the microcontroller can be switched to a low-power mode[69] while not running the algorithm. This mode allows for a static 10.8 μW power consumption and has a wake-up time of 113 μs. Considering the clock rate of 84 MHz, the microcontroller terminates all the operation of the neuromorphic algorithm well within the 10 ms period, with a 6.3% duty cycle for the algorithm computation, thus taking advantage of the low-power mode. The resulting power consumption is 244.7 μW. Note that we omit the derivation of the ITD from the ToFs and the conversion to the detected angle, thus underestimating the power consumption in the microcontroller. This gives further value to the energy efficiency of the proposed system. As a further term of comparison, we estimate the power consumption of a classical beamforming approach, presented in refs. [31,54], when embedded on the same microcontroller[68] under a 1.8 V supply voltage. Five evenly spaced pMUT membranes are used to provide data for the beamforming. About the processing itself, the beamforming technique used is the delay-and-sum. It simply consists of applying a delay to the channels corresponding to the expected time difference of arrival between one channel and a reference channel. If the signals are in phase, once time-shifted, the sum of these signals will exhibit high energy. If they are not in phase, destructive interference will limit the energy of their sum. In ref. [31], a 2 MHz sample rate is chosen to time-shift the data by an integer number of samples. A more frugal approach consists in keeping a coarser 250 kHz sample rate and using finite-impulse-response (FIR) filters to synthesize fractional delays. We will consider that the beamforming algorithm complexity is dominated by the time-shifting because of the convolution of each channel with a 16 taps FIR filter for each direction. To calculate the number of MIPS required for this operation, we consider using a 6-ms window per measurement to capture a 1-meter range, 5 channels, 11 beam-forming directions (+/−50° range with a 10° step). Already 75 measurements per second push the microcontroller to its maximum of 100 MIPS. According to ref. [68], this results in a power consumption of 11.26 mW, which gives a total power consumption of 11.71 mW when adding the on-chip ADC contribution.

## Data availability
The data that support the findings of this study are available from the corresponding author, F.M., upon reasonable request.

## Code availability
All software programs used in the presentation of the article are freely available upon request.

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

## Acknowledgements
We acknowledge funding support from the H2020 MeM-Scales project (871371). This work is also supported by the ERC DIVERSE project (101043854) as well as the French ANR via Carnot funding. In addition, we thank J.-F. Nodin, F. Andrieu, S. Bonnetier, T. Hirtzlin (CEA-Leti), D. Querlioz (Université Paris-Saclay, CNRS), and A. Valentian (CEA-List) for discussing various aspects of the article.

## Author contributions
F.M., T.D., M.P., and E.V. developed the idea of processing pMUT sensory input with neuromorphic RRAM-based circuits. F.M., E.H., B.F., F.B., F.G., T.M., and F.R. contributed to the measurement and analysis of the pMUT sensory system. P.C. and J.C. provided insights into the anatomy of the barn owl's auditory system and proposed ideas to implement the biological concepts in circuits. T.D. and M.P. designed and laid out the RRAM-based circuits, and F.M. and A.D.P. characterized them. E.E. and N.C. performed the measurements on the 16-kb RRAM array. E.V. directed the work. M.P., G.I., and D.E. participated in the discussion and provided guidance to the project. All authors contributed to writing the paper.

## Competing interests
The authors declare no competing interests.
