## [Peer Review File · Nature Communications]

Title: Neuromorphic object localization using resistive memories and ultrasonic transducersREVIEWER COMMENTS

Reviewer #1 (Remarks to the Author):

This work presents results on the potential of hardware neuromorphic systems, using memristive devices, on low-power, computational performance, and low latency capabilities on mimicking a challenging biological system (hearing system of a barn owl). The paper novelty is under no discussion. The topic is of interest for the field and it deserve for publication.

The main concern from this reviewer is the fact that barn owl hearing system works in the range 1-8kHz, but this work uses ultrasound devices that work in the range of 110-117kHz. In this sense the proposed system is not really mimicking the owl. It is closer to a bat, which produce and detect ultrasounds in the range of up to 200kHz in some species. This must be justified / corrected in the paper.

Some other minor concerns:

- Missing reference related to ITD implementation: there is a recent FPGA implementation that authors should review and check if it is related to the work presented here:

<https://doi.org/10.1109/TNNLS.2021.3108047>

- Authors claim that they propose a full neuromorphic system, so a justification on why pMUT are neuromorphic should be clearer.

- In page 5 authors said that they "simply generate a spike at the reception of the sound wave". Can you clarify the nature of this sound and if it is just one spike of a stream of them?

- In page 6 authors refer to the Drosophila's visual system for justifying the use of the direction-sensitive CD because of its similarity to the EMD. Since visual and sound information are different, shouldn't you justify the use of one circuit from one system to the other?

- Power estimation for the microcontroller: Can you offer the real power consumption (static and dynamic) on both implementations (microcontroller and RRAM)? It is not clear from where your estimations are coming. Are they taken from the datasheet of the microcontroller?

Reviewer #2 (Remarks to the Author):

The manuscript by Moro et al. presents an interesting bio-mimetic, event-driven sensory processing system for a real-world object localization application. The system integrates a full-custom piezoelectric micromachined ultrasound transducer (pMUT) sensors with neuromorphic RRAM-based coincidence detectors and delay line circuits. The system-level simulations, which are calibrated by experimental data, show encouraging results including orders of magnitudes higher energy efficiency than a

microcontroller. The key idea was proposed in ref. 44 (Jeffress model), and to some extent, this work implements it in hardware. Overall the paper is nicely written. There are several major concerns that need to be addressed before it can be considered for publication on Nature Communications:

1. First of all, as one of the innovations of this work, it is necessary to provide the experimental results of RRAM-based delay line and coincidence detector, instead of SPICE simulations in Fig. 4c, d, e.

2. “We present measurement results from the fabricated system comprising resistive memories-based coincidence detectors, delay line circuits, and a full-custom pMUT sensor. We use these experimental results to calibrate our system-level simulations”. The authors should discuss more on how to assemble those components into a fully functional system, and whether any interface circuitry is needed. For example, how to connect the signal output of LIF neurons to input of RRAM-based neuromorphic computational map?

3. Following the question above, the system-level assessment hence seems to be incomplete and does not take into consideration all the necessary components in the functional system demonstration. The conversion of ITD to the angle value is neglected. What is the contribution of this part in the system performance evaluation, such as the energy efficiency?

4. The experiment results also need to be clarified whether they are performed on a 16kb array or a neuromorphic platform. How many devices were actually used?

5. In the introduction part, the authors state that “In this work we propose to exploit the ability of RRAMs to change their conductance value in a non-volatile manner to counteract the variability in analog circuits”. However, it is known that the conductance states of RRAM devices could fluctuate and drift over time. It is recommended to discuss on the non-ideal effect of RRAM, e.g., relaxation/retention, noise, limited precision, on the localization results.

6. The basic characteristics of RRAM devices and also the pMUT sensors should be provided since they are the key components in the proposed neuromorphic system. Also, the challenge of coupling the pMUT and a neuromorphic resistive memories-based computational map is not stated clearly in the introduction.

7. Some of the results do not seem to be consistent with each other in this paper. For example, Fig. 2f shows the angle precision with different object distances when the object angle is 0° . The values are inconsistent with the results of Fig. 2e (e.g., angle precisions of $\sim 25^\circ$ in Fig. 2f vs $\sim 15^\circ$ in Fig. 2e for $D = 60\text{cm}$).

8. There have been many memristor-based and CMOS sound localization works with ITD cues. It is suggested to cite them and compare them with the proposed approach in this work.

9. Also, in the previous references (B. Grothe, *Frontiers in Neural Circuits*, 2014; B. Razavi, 2nd

International IEEE EMBS Conference on Neural Engineering, 2005), ITD cues seem to be more effective for frequencies below 2kHz. It is different from the frequency bands in this work. Does this affect the detection accuracy?

10. Please clarify the detection range in this work, and comment on the challenges of expanding to a wider range.

11. In Fig. 6d, it appears that the sum of false positive rate (i.e. rate of events correctly detected as correlated) and true positive rate (i.e. rate of events incorrectly detected as correlated) is not 1. Why is that?

12. Minor comments:

1) Several abbreviations repeated multiple times, such as pMUT.

2) Some of the references are not correctly listed, e.g., refs. 40, 49, 60.

Response to the reviewers: Neuromorphic object localization using resistive memories and ultrasonic transducers

We sincerely thank all the reviewers for their insightful comments on our manuscript. We believe addressing the comments has made the paper much stronger and we are grateful for that. Below, we reply to each concern of the reviewers one by one. We have addressed the changes in the main and supplementary texts, highlighted with a strike-through red when text was removed, and in blue when text was added.

Response to Reviewer 1

This work presents results on the potential of hardware neuromorphic systems, using memristive devices, on low-power, computational performance, and low latency capabilities on mimicking a challenging biological system (hearing system of a barn owl). The paper novelty is under no discussion. The topic is of interest for the field and it deserve for publication.

We thank the reviewer for acknowledging the novelty of our work and its importance for the field of neuromorphic engineering.

The main concern from this reviewer is the fact that barn owl hearing system works in the range 1-8kHz, but this work uses ultrasound devices that work in the range of 110-117kHz. In this sense the proposed system is not really mimicking the owl. It is closer to a bat, which produce and detect ultrasounds in the range of up to 200kHz in some species. This must be justified / corrected in the paper.

Reply 1

We thank the reviewer for bringing up this point. We fully understand the rational of their comment concerning the frequency range. We agree that the sensory system in our work diverges from that of the owl and it is closer to the system bats use for echo-localization. To justify our choices, we want to make clear that our goal was to conceive an object localization system inspired by biology, not to build a bio-mimetic system. We have been inspired by the Jeffress Model as a computing strategy to localize auditory cues based on ITDs. The topographic wiring used in this work comes from the barn owl. Not only is there ample work and anatomical evidence of the presence of Jeffress-like mechanism in owls^{1,2}, the existence of this mechanism is actually unclear in bats, which may not even use time delays in the same way³. Thus, we kept the barn owl's scheme of information processing and used a bio-inspired (not bio-mimetic) approach in allowing ourselves to change the range of frequencies used.

The choice of an ultrasound sensor was pondered around engineering and optimization criteria. Firstly, restricting the band of reception to - ideally - a single frequency improves the accuracy of the measurement and simplifies the post-processing stage. Additionally, operating in the ultra-sound regime has the advantage of making the emitted pulse inaudible, and thus not bothersome for humans as their auditory range is approximately [20-20k] Hz. Consequently, we developed pMUTs working at around 117 kHz, with diameter size of 880 μm , proven to be reliable and efficient. This technology provides us with a highly miniaturized, energy efficient, narrow-band sensor operating in the ultra-sound regime, which is greatly desirable for the object localization task. We hope that the present discussion can clarify and support our choices about the sensory system. We took the following actions to clarify our choices in the main text based on the reviewer's comments.

Action 1: We underlined that our system differs from the barn owl hearing system in terms of operating frequency and clarified the choice of the ultrasound sensors in the main text, at lines [76-82].

Action 2: We replaced the term *bio-mimetic* with *bio-inspired* in the main text.

Missing reference related to ITD implementation: there is a recent FPGA implementation that authors should review and check if it is related to the work presented here: <https://doi.org/10.1109/TNNLS.2021.3108047>

Reply 2 We thank the Reviewer for pointing out this work⁴, which proposes a digital implementation of the Time-Difference-Encoder (TDE), representing an alternative to the Jeffress model implemented in our work. It is a relevant reference for our paper. Furthermore, this work establishes a benchmark for the object localization system power consumption. The measured power consumption of a TDE unit implemented on a FPGA was reported to be approximately 1.5 mW. Because FPGAs are not optimized for power and follow Von-Neumann architecture, they burn a lot more power compared to equivalent Application-Specific-Integrated-Circuits (ASIC) implementations. The simulated power for the ASIC implementation of the TDE is 1.4 nW static and 500 μW in operation mode at 500 kHz. These values have been calculated for a single TDE unit, while the whole localization operation requires multiple TDE modules. The total system power consumption of the analog implementation of the Jeffress model proposed in our work is 81.6 nW. This confirms the energy efficiency of the proposed

bio-inspired RRAM-based system (see Fig. R2).

Action 1: We have added the missing reference in the main text, in the introduction section, line 40.

Action 2: We have added the power estimations of the TDE unit implemented on FPGA calculated in reference⁴ for comparison with our system in the main text lines [259-261] and in Fig. 7b of the main text .

Authors claim that they propose a full neuromorphic system, so a justification on why pMUT are neuromorphic should be clearer.

Reply 3 We recognize that pMUT sensors have not been developed with the aim of mimicking a biological sensing apparatus, and cannot be considered as neuromorphic per-se. However, we claim that the utilization we made of pMUT, leveraging the pre-processing stage described in Figs. 2b and c in the main text, is inspired by biology. This is because analog sensory information is directly translated into a train of spikes without any need for storing the sound wave into memory (for more details see Reply 4). This feature is shared with systems such as the Silicon Cochlea⁵ and NAS⁶, representing examples of neuromorphic sensory systems. Based on these arguments, we believe that our system can be considered as neuromorphic.

Action: We modified the main text (lines 87-88) to clarify that although the pMUT sensory system is not strictly bio-inspired per-se, the utilization we made of it is inspired by biology.

In page 5 authors said that they "simply generate a spike at the reception of the sound wave". Can you clarify the nature of this sound and if it is just one spike of a stream of them?

Reply 4 Thanks to this comment we realized that the neuromorphic pre-processing system can be simplified compared to the one proposed in the original version of the manuscript. The Integrate-and-Fire neuron circuit can be removed, allowing to further simplify the system and reduce the power consumption.

The neuromorphic pre-processing system converts the information of the received sound wave to the timing of a single spike. To understand how this is achieved, we go through the details of our proposed neuromorphic pre-processing of the sound information below. Fig. R1 reports all the stages and corresponding hardware components of the adopted neuromorphic pre-processing technique converting the two sound waves detected by the two receiving pMUTs into input spikes for the neuromorphic computational map. The same procedure is adopted for the two pMUT receivers. The reflected sound wave reaches the two pMUT receivers in a slightly different time, depending on the azimuthal position of the object. The frequency of the reflected sound wave is approximately the same as the sound wave produced by the emitter pMUT and the wave peak amplitude corresponds to the Time-of-Flight (ToF). The ToF is extracted from each of the two channels and encoded as the precise timing of a single spike. The receiving pMUTs convert the reflected sound wave into a voltage signal that is first magnified by a pre-amplifier and then smoothed by a Band-Pass filter centered around the pMUTs oscillation frequency. The resulting signal is Half-Wave rectified to remove the negative component and then fed into a Leaky Integrate-and-Fire (LIF) neuron circuit. The threshold of the LIF is calibrated to obtain a single output spike at the ToF of the reflected sound wave detected by the pMUT receiver. Since the amplitude of the pMUT response decays as a function of the object distance, the threshold of the LIF neuron must decay over time: an exponentially decaying threshold correctly reproduces the behavior of the decrease in peak amplitude.

Action 1: We added the Supplementary Note *pMUT sound reception and pre-processing stage* and Figure S4 showing all the stages and corresponding hardware components of the adopted pre-processing technique converting the two pMUT input sound waves into input spikes for the neuromorphic computational map.

Action 2: Added a sentence to clarify that a single spike is generated per received sound wave (corresponding to the ToF), lines [101-104] in the main text.

Action 3: We have modified Fig. 2c to present the the new pre-processing system without the IF neuron circuit. The single bank power consumption has been updated line 244 in the main text.

Action 4: We have updated Figs. 2d, e and f, and 7 taking into account the new pre-processing system without the IF neuron circuit.

In page 6 authors refer to the Drosophila's visual system for justifying the use of the direction-sensitive CD because of its similarity to the EMD. Since visual and sound information are different, shouldn't you justify the use of one circuit from one system to the other?

Reply 5 Coincidence Detection between spatially and temporally distributed events is a basic operation common to a wide

Figure R1. Schematic of the complete object localization system. From left to right, the pMUT raw signal is first magnified by a pre-amplifier. A Band-Pass Filter (BPF) centered around the pMUT resonance frequency is applied to remove noise and smoothed the signal. The resulting signal is Half-Wave-Rectified to remove the negative component and fed into an Integrate and Fire (IF) neuron circuit, which accumulates the signal over time in its membrane voltage and emits a spike in output when overcoming a predetermined threshold potential. The frequency of the output spike train is proportional to the amplitude of the input signal. A Leaky-Integrate-and-Fire (LIF) neuron accumulates the spike train coming out of the IF neuron. This LIF neuron features a threshold that decreases exponentially in time which, once overcome, induces a single output spike emitted at the Time-of-Flight of the sound wave received by the pMUT.

range of sensing tasks, such as optical flow-based obstacle avoidance⁷ (i.e. the EDM system), and sound source localization⁸. Consequently, the computation of both direction sensitive and insensitive CDs are fundamental building blocks to build both visual and sound localisation systems.

The reason why a direction sensitive CD is useful in vision is that it efficiently allows to detect the direction of motion of objects in space. This is of vital importance for flies, allowing them to avoid collisions with objects and guide through tight spaces^{9,10}. This principle of motion detection could for example be of great use for autonomous neuromorphic robots, reducing the cost of processing visual stimuli⁷. A possible issue related to the use of the proposed neuromorphic computational map for visual applications is that the range of required time scales may be different from the one of the sound system. As explained in the Supplementary Material, the proposed implementation of the circuit fits a range of 4 orders of magnitude of time scales, as shown by the characterization of the time constant. This makes the Neuromorphic circuit platform fit in different tasks characterized by different temporal features.

Action: We added few sentences to clarify this point in the main text lines [181-186].

Power estimation for the microcontroller: Can you offer the real power consumption (static and dynamic) on both implementations (microcontroller and RRAM)? It is not clear from where your estimations are coming. Are they taken from the datasheet of the microcontroller?

Reply 6

Thank you for your comment. Zero static power consumption is among the advantages of analog RRAM-based computation compared to conventional digital electronics. Due to the use of non-volatile memory, the circuit has an instant on/off feature: it

Figure R2. Power consumption of the presented neuromorphic sensory and signal processing system. The power consumption of the proposed system is compared with an implementation of an equivalent neuromorphic algorithm and of a Beamforming system on a STM32 microcontroller, as well as a Temporal Difference Element circuit implemented on a FPGA⁴. The proposed system has zero static consumption, while the microcontroller in Low-Power-Mode consumes $10.8 \mu\text{W}$. The TDE circuit does not perform object localization, but it can be employed in a Jeffress-like model to retrieve the object's angle based on the ITD information.

works immediately after being turned on, allowing to cut the power supply entirely as soon as the circuit is not used. In the power consumption assessment, we thus considered that the neuromorphic computational map is switched off (i.e. zero static power) when no information is to be processed. The reported 81.6 nW corresponds to dynamic power consumption and it has been calculated considering a rate of localization operations of 100 Hz (corresponding to 10 ms for localization operation) and an activation period of $300 \mu\text{s}$ (maximum possible ITD).

For what concerns the micro-controller, we referred to its Data-sheet¹¹ to calculate power consumption. We took into account the energy cost for all the operations required to compute the proposed object localization task. Stimulated by the insightful comment of the reviewer, we discovered a particular version of the STM32¹¹ microcontroller platform, called STM32F401, which offers a Low-Power-Mode¹². In this mode, the SRAM cells are maintained under a voltage of 1.2 V , conserving the necessary parameters of the algorithm, while the clock is disengaged. This Low-Power-Mode allows to reduce the static power consumption to $10.8 \mu\text{W}$. The wake-up time to shift to the normal operation is $113 \mu\text{s}$. The microprocessor can be switched to this Low-Power-Mode while not performing the algorithm's operations.

In our case, we have chosen to separate each cycle of the pMUT emission by 10 ms , which corresponds to a localization operation. This determines the *time window* for the computation to be completed. Considering the clock rate of 84 MHz , the microprocessor terminates all the operations of the neuromorphic algorithm on the STM32 in about $630 \mu\text{s}$. Therefore, for the remaining of the 10 ms time window, the processor can be switched to the Low-Power-Mode. Instead, the Beamforming algorithm requires the full 10 ms to run on the microcontroller. Therefore, the microcontroller is always in active mode when running the Beamforming algorithm, and it cannot benefit from the Low-Power-Mode. Therefore, the resulting power consumption for running the Neuromorphic algorithm is reduced from $991 \mu\text{W}$ to $244.7 \mu\text{W}$ ($233.9 \mu\text{W}$ dynamic and $10.8 \mu\text{W}$ static power consumption), while the power consumption of the Beamforming implementation is unchanged. The updated power consumption values are reported in Fig. R2

Action 1: We modified Fig. 7b in the main text.

Action 2: We updated the power consumption estimation in the Method section, we have now specified the static and dynamic contributions for the Neuromorphic approach calculated thanks to the Low-Power-Mode, lines [401-406].

Action 3: We added a sentence in the main text (lines [136-137]) to underline that the proposed solution has zero static power consumption thanks to the non-volatility of RRAM devices.

Response to Reviewer 2

The manuscript by Moro et al. presents an interesting bio-mimetic, event-driven sensory processing system for a real-world object localization application. The system integrates a full-custom piezoelectric micromachined ultrasound transducer (pMUT) sensors with neuromorphic RRAM-based coincidence detectors and delay line circuits. The system-level simulations, which are calibrated by experimental data, show encouraging results including orders of magnitudes higher energy efficiency than a microcontroller. The key idea was proposed in ref. 44 (Jeffress model), and to some extent, this work implements it in hardware. Overall the paper is nicely written. There are several major concerns that need to be addressed before it can be considered for publication on Nature Communications:

Thank you for your encouraging comments about our work.

1. First of all, as one of the innovations of this work, it is necessary to provide the experimental results of RRAM-based delay line and coincidence detector, instead of SPICE simulations in Fig. 4c, d, e.

Reply 7 We agree with the Reviewer, we have replaced the SPICE simulations with experimental results to show the behavior of the neuromorphic platform circuit in its three configurations: delay line, direction insensitive CD, and direction sensitive CD. Experimental measurements are plotted in Fig. R3. We are confident that these measurements can clarify the behavior of the circuit.

Action: We have replaced the SPICE simulations with experimental results in Fig. 4.

Figure R3. Experimental measurements of the neuromorphic circuit platform. (Delay Line) The input spike presented at input V_{in} charges the membrane voltage of the neuron N_1 . Upon reaching of the membrane threshold, the output spike is emitted with a delay T_{del} . (Direction Insensitive CD) The experiment is set presenting 2 different delays, within and outside the detection range, producing an output spike only when the input spikes are coming in temporal proximity. (Direction Sensitive CD) The circuit responds when the inputs are presented with temporal proximity and with a certain order (in this example input 0 before input 1). The output of the circuit is represented by neuron 1 (N_1).

2. “We present measurement results from the fabricated system comprising resistive memories-based coincidence detectors, delay line circuits, and a full-custom pMUT sensor. We use these experimental results to calibrate our system-level simulations”. The authors should discuss more on how to assemble those components into a fully functional system, and whether any interface circuitry is needed. For example, how to connect the signal output of LIF neurons to input of RRAM-based neuromorphic computational map?

Reply 8 This comment represents a very stimulating input and allows us to expand on what was mentioned in the main text. We remark that the simplicity of the proposed solution allows us to minimize the required circuitry and interface blocks. All the computation is performed in the analog domain, and the components of the pre-processing (Fig. 2c) and the neuromorphic computational map (based on the circuit in Fig. 4a) are fully compatible. Co-integration of both parts in the same chip is therefore foreseeable in the future. Thanks to this comment we realized that the neuromorphic pre-processing system can be

simplified compared to the one proposed in the original version of the manuscript. The Integrate-and-Fire neuron circuit can be removed, allowing to further reduce the power consumption.

To clarify the proposed solution, we prepared a schematic of the complete object localization system (Fig. R1 of this document) presenting all the circuits required to connect the pMUT raw signal to the input of the RRAM-based neuromorphic computational map. The receiving pMUT converts the reflected sound wave into a voltage signal that is first magnified by a pre-amplifier and then smoothed by a Band-Pass filter centered around the pMUT's oscillation frequency. The resulting signal is Half-Wave rectified to remove the negative component and then fed into a Leaky-and-Integrate-and-Fire (LIF) neuron circuit. The threshold of the LIF is calibrated to obtain a single output spike at the ToF of the reflected sound wave detected by the pMUT receiver. Since the amplitude of the pMUT response decays as a function of the object distance, the threshold of the LIF neuron must decay over time: an exponentially decaying threshold correctly reproduces the behavior of the decrease in peak amplitude. An up-level-shifter ensures that the neuromorphic pre-processing and the neuromorphic computational map, operating at different voltages, operate correctly when connected together. The output spikes are then directly sent to the delay lines and coincident detector modules organized in parallel into the neuromorphic computational map. Since the spikes are sent to the gate of the RRAM access transistors no additional amplification circuitry is required.

Action 1: We have added Supplementary Figure S4 presenting a schematic of the complete object localization system including all the circuits required to connect the pMUT raw signal to the input of the RRAM-based neuromorphic computational map.

Action 2: Added a sentence in the main text (line [109-113]) to underline that the the signal output of the LIF neurons are directly sent to the neuromorphic computational map.

Action 3: We have modified Fig. 2c in the main text to show that the output spikes generated by the neuromorphic pre-processing system are directly sent to the delay lines and coincident detector modules organized in parallel into the neuromorphic computational map, and to present the new pre-processing system without the IF neuron circuit. The single bank power consumption has been updated in line 243 in the main text.

Action 4: We have updated Figs. 2d, e and f and 7 taking into account the new pre-processing system without the IF neuron circuit.

3. Following the question above, the system-level assessment hence seems to be incomplete and does not take into consideration all the necessary components in the functional system demonstration. The conversion of ITD to the angle value is neglected. What is the contribution of this part in the system performance evaluation, such as the energy efficiency?

Reply 9 In the Jeffress model, the location of an object is computed based on ITD clues and encoded as the activation - i.e. output spike - of a coincidence detector neuron in the computational map. This means that there is a univocal relation between each CD neuron and a particular azimuthal location in space. As a consequence, the activation of a particular neuron encodes the position of the target object in space and there is no need to convert the ITD to the computed angle (see Fig. R4). The output of the computational map is already in the spike domain and it can be used as an input for a spike-based neuromorphic processor for further computations, for example to make decisions based on different sensors data fusion.

Action 1: We added a sentence in the main text (line [63-64]) to clarify that the activation of a particular neuron in the neuromorphic computational map directly encodes the position of the target object in space and there is no need to further convert the ITD into an angle.

Action 2: We modified Fig. 2c in the main text to clarify that there is a univocal relation between each CD neuron and the output angle.

4. The experiment results also need to be clarified whether they are performed on a 16kb array or a neuromorphic platform. How many devices were actually used?

Reply 10 We thank the reviewer for this comment. Below we have made a list of figures containing experimental measurements on RRAM devices and for each one we have indicated if they have been performed on the 16 kb array or on the neuromorphic platform and we have specified on how many devices the measurement was carried on. The 16 kb RRAM array and the neuromorphic circuits have been fabricated and tested on the same wafer.

Fig. 3c of the revised manuscript presents the statistical characterization of the RRAM technology on a 16 kbit array. By adjusting the programming conditions (programming current) the RRAM cells conductances can be modulated in different levels. Figs. 3d and e present experimental measurements of the fabricated basic building blocks of the proposed neuromorphic computational map, with N 1T1R structures connected to a DPI synapse and a LIF neuron (Fig. 3b). Measurements have been performed on one circuit and demonstrate the dual function of the RRAMs, which can both route and weigh the input pulses. Figs. 4 and 5 present experimental measurements of the fabricated RRAM neuromorphic platform depicted in Fig. 3a.

Figure R4. Jeffress Model implementing the conversion between ITD and object azimuthal location. (a) Simple scheme illustrating the link between the angular position of the target object, its ITD and the location of the Coincidence Detector neurons in the computational map. (b) The output of the system is encoded as the activation - i.e. the output spike - of the CD neuron corresponding to the object position. In this case, the CD corresponding to an angular position of -30° produces an output spike (in red).

The measurements presented in Fig. 4 have been performed on a single circuit and demonstrate that the proposed circuit can assume a delay line configuration, as well as two distinct coincidence detectors functionalities. Fig. 5 is intended to present circuit-to-circuit variability in analog neuromorphic systems. Fig. 5a presents the response of 9 neurons measured on 9 different neuron circuits. Figs. 5b and c present the impact of the neuron to neuron variability on the direction-sensitive CD. Measurements have been performed on the same circuit, under different input conditions. The non-ideal effects of RRAM (i.e. conductance variability and relaxation/retention) have been estimated thanks to extensive electrical characterization of the 16 bk array. The results are presented in Fig. S1 in the Supplementary Information. Those results were then employed to calibrate the system level simulations reported in Figs. 6 and 7.

Action: We have re-written the captions of Figs. 3, 4 and 5 specifying if they have been performed on the 16 kb array or on the neuromorphic platform and we have added on how many circuits and devices the measurement was carried on.

5. In the introduction part, the authors state that “In this work we propose to exploit the ability of RRAMs to change their conductance value in a non-volatile manner to counteract the variability in analog circuits”. However, it is known that the conductance states of RRAM devices could fluctuate and drift over time. It is recommended to discuss on the non-ideal effect of RRAM, e.g. relaxation/retention, noise, limited precision, on the localization results.

Reply 11 It is true that the conductance state of resistive memory devices could fluctuate and drift over time. This issue is an important point that we did not include in the original version of the paper. To address this aspect we proposed a new RRAM programming technique validated through additional experiments. To overcome the conductance drift effect, we proposed to add a wait time of 5 s after each RRAM programming operation of the classical iterative programming strategy¹³. Cells that suffer from conductance instability during the waiting period are rescheduled to the next programming iteration, allowing the algorithm to take into account and correct the conductance drift. Fig. R5 shows the a conductance level programmed with the conventional iterative programming strategy and with the new programming method featuring the relaxation correction. For the devices programmed with the conventional strategy the conductance relaxation/retention causes the initial distribution to spread over time, and after one hour, more than 45% of the programmed devices are out of the target conductance range. For the new programming strategy with relaxation correction, the effect of conductance relaxation/retention is negligible after one hour (less than 6% of cells are out of the target range). Therefore, this technique should be adopted to correctly program the RRAM cells of neuromorphic computational map. The 5 s waiting time has been added in the Algorithm 1, Delay Line RRAM calibration, and Algorithm 2, Direction Insensitive CD calibration, reported in the Supplementary Material.

The new programming scheme with relaxation correction needs more iterations to achieve the same programming performance compared to the classical iterative programming one. Consequently, the number of iterations to calibrate the delay line and the

CD circuits increases. Figs. S1d and e show the impact of the new programming technique with relaxation correction on the number of iterations required to calibrate the delay lines and the direction insensitive CD circuits.

Action 1: We have added Supplementary Note *RRAM characterization* and Figure S1 demonstrating the effectiveness of the new smart programming technique with relaxation correction.

Action 2: We have updated Supplementary Note *RRAM Calibration Procedure in delay line and direction insensitive coincidence detector* to adopt the programming strategy with relaxation correction to correctly program the RRAM in the delay line and direction insensitive coincidence detector.

Action 3: We have corrected Fig. 6 in the main text by duly taking into account the impact of the new programming technique with relaxation correction on the number of iterations.

Figure R5. (a) Conductance distributions of 1096 cells programmed with the standard smart programming technique and the new one with relaxation correction read just after programming ($t=0$) and after 1 hour ($t=1\text{H}$). (b) visualizes the same data as the PDF of the conductance values at $t=1\text{h}$ ($G_{t=1h}$) minus the conductance value after programming ($G_{t=0s}$).

6. The basic characteristics of RRAM devices and also the pMUT sensors should be provided since they are the key components in the proposed neuromorphic system. Also, the challenge of coupling the pMUT and a neuromorphic resistive memories-based computational map is not stated clearly in the introduction.

Reply 12 To provide the basic characteristics of RRAM cells we added a paragraph and Figure S1 in the Supplementary Materials *RRAM characterization*. Moreover, we complemented the *Acoustic measurement setup and pMUT characterization* section in the Methods providing more details on the electrical characterization of the adopted pMUT sensors.

The pMUT sensors have reached industrial maturity. However, the promising possibilities of pMUTs as multi-membranes, angular-resolved sensors have barely been exploited, most probably because the signal processing required to extract advanced information does not suit low-power sensing. We believe that the proposed system coupling pMUT sensors and the RRAM based neuromorphic computational map can overcome this challenge. In addition, the event-driven approach allows for maximum compression of acoustic information detected by the pMUT (two spikes for a single object localization operation) and satisfies the requirements of the computational map, which is event-based. This strongly reduces the interface circuits required between the pMUT sensors and the neuromorphic computational map.

Action 1: We have added a paragraph and Figure S1 in Supplementary Material describing the basic electrical characteristics of RRAM cells.

Action 2: We have complemented the *Acoustic measurement setup and pMUT characterization* section in the Methods providing more details on the electrical characterization of the adopted pMUT sensors.

Action 3: We have added lines [92-95] in the main text underling the main challenge of the coupling pMUT sensors and the neuromorphic computational map.

7. Some of the results do not seem to be consistent with each other in this paper. For example, Fig. 2f shows the angle precision with different object distances when the object angle is 0° . The values are inconsistent with the results of Fig. 2e (e.g. angle

precisions of 25° in Fig. 2f vs 15° in Fig. 2e for $D = 60\text{cm}$).

Reply 13 We apologise for the error in Fig. 2. The results of the pre-processing stage depend on some parameters, such as the IF neuron threshold and the LIF leakage rate. We mistakenly used slightly different values of the latter in Fig. 2e and f. The Figures have been corrected.

Action: We have corrected the data in Figs. 2e and f.

8. There have been many memristor-based and CMOS sound localization works with ITD cues. It is suggested to cite them and compare them with the proposed approach in this work.

Reply 14 Thanks for the comment, which made us realize that some important references were missing in the original manuscript, that have now been added in the revised paper. The first important reference has been suggested by reviewer 1 and proposes a digital implementation of the Time-Difference-Encoder (TDE) model, representing an alternative to the Jefferss model⁴. This work implemented the digital TDE model on an FPGA. The measured power consumption is approximately 1.5 mW. The simulated power for the ASIC implementation is 1.4 nW static and 500 μW in operation mode at 500 kHz. These values have been calculated for a single TDE unit, while the whole localization operation requires multiple TDE modules. The total system's power consumption of the analog implementation of the Jefferss model proposed in our work is 81.6 nW. This confirms the energy efficiency of the proposed bio-inspired RRAM-based system (see Fig. R2).

We added a second important reference proposing a fully connected neural network composed of two input and two output neurons connected via 2x2 RRAM array for sound localisation by ITD detection¹⁴. This work emphasizes the effectiveness of using RRAM based analog circuit for sensory processing. A quantitative comparison with our system is difficult since the authors have not implemented the full localization system, but have encoded the ITD in the form of a voltage. To perform object localization, that voltage has to be converted to the object's angular position. This paper implements Spike Timing Dependent Plasticity (STDP) to define the synaptic weight. This technique is a valid alternative to the calibration procedure proposed in our work to mitigate the impact of the variability of analog circuits.

In the revised paper we have also added a citation to additional works related to either memristor- or CMOS-based neuromorphic circuits to perform either ITD detection or object localization¹⁵⁻¹⁷.

Action 1: We added refs^{4, 14-17}.

Action 2: We have added the power estimations of the TDE unit implemented on FPGA calculated in reference⁴ in Fig. 7b in the main text.

Action 3: We have added lines [208-209] in the main text to underline that STDP is an alternative to our calibration procedure.

*9. Also, in the previous references (B. Grothe, *Frontiers in Neural Circuits*, 2014; B. Razavi, *2nd International IEEE EMBS Conference on Neural Engineering*, 2005), ITD cues seem to be more effective for frequencies below 2kHz. It is different from the frequency bands in this work. Does this affect the detection accuracy?*

Reply 15 When operating at 2 kHz, the ITD can be detected by phase-locking, exploiting the difference in phase of the sound waves at the 2 ears. This is what many animals, such as the barn owl, do. The [1-3] KHz frequency range is particularly suited to capture ITDs by phase-locking when the sound receptors (ears) are located at a distance d of [2-15] cm from each other. This is because the placement of the ears translates to ITDs in the order of 100 μs , within the period of the sound wave. Instead, when operating in the ultrasound regime (>100 kHz), phase-locking requires precision of the order of 10 ns, which is too fast for biological system and neuromorphic circuits. We thus use the peak amplitude (i.e. the Time of Flight) of the two sound waves at the 2 pMUT receivers to compute the ITD. The method - described in the main text lines [103-114] and in the Supplementary Material Fig. S4- can approximately be summarized as the demodulation of the signal's amplitude envelope from the pMUT raw signal. This method poses a lower bound for accuracy: the smallest ITD we can detect is roughly equal to one period of the signal we analyze.

$$ITD_{min} \approx 1/f_{pMUT, resonance\ frequency} \quad (1)$$

Since the resonance frequency of the pMUT sensors used in this work spreads in the range [110 - 112] kHz the lower bound for accuracy is about 10 μs , corresponding to an angular resolution of approximately 4°.

Action 1: We have modified Fig. 2 e in the main text to show the minimum angular resolution of 4°. We have mentioned in the caption that this resolution limit is set by the peak amplitude method employed in this work, which in turn results in a 1 μs

best accuracy for the ITD detection.

Action 2: We clarified the choice of working at ultrasound frequencies in the main text at lines [77-82].

10. Please clarify the detection range in this work, and comment on the challenges of expanding to a wider range.

Reply 16

The detection range depends on the pMUT's performances, both as emitter and as receiver. Focusing on emission, the air-pressure produced by the emitter greatly varies the detection range. The pressure sent by the pMUT may be increased by raising up the voltage driving the pMUT's membrane, as discussed in¹⁸. The authors of the above reference show that a careful calibration of the membrane driving voltage allowed to maintain a 12 dB Signal-to-Noise ratio in a setup where the emitter and receiver are separated by 2 m. Please notice that a face to face placement of the emitter and receiver does not represent a realistic scenario for the utilization of pMUTs, but it is a popular characterization setup. Another option in order to increase the detection range is to combine several membranes within one single emitter, as demonstrated in¹⁹, where it is claimed that a 2x2 pMUT array yields a "10x improvement in pressure output". Furthermore, using other designs for the pMUT membrane, the resonant frequency could be shifted to lower values: this would increase the detection range since absorption is lower at lower frequencies.

Other improvements might be performed on the receiving side. The noise floor of the pMUT receiver could be drastically reduced improving the connections between the pMUT and the first stage amplifier, currently achieved by wire-bonding and RJ45 cables. These design improvements are expected to push to longer distances the degradation of the angular resolution, that is currently observed at a distance of about 60 cm.

Action: This is now clarified in the main text lines [122-127].

11. In Fig. 6d, it appears that the sum of false positive rate (i.e. rate of events correctly detected as correlated) and true positive rate (i.e. rate of events incorrectly detected as correlated) is not 1. Why is that?

Reply 17 We defined the True Positive (TP) events as the number of Coincidence Detectors circuits that correctly respond to temporally correlated inputs. For t_i and t_j corresponding to the input spike times and Δt_{CD} the target coincidence range, the CD has to respond if

$$|t_i - t_j| < \Delta t_{CD} \quad (2)$$

and should not respond otherwise. We defined the False Positive (FP) events as the number of CD circuits that respond when the previous equation is not satisfied. False Negative (FN) events occur when Equation (1) is satisfied but the CD does not activate. The True Positive Rate (TPR, also called sensitivity), False Positive Rate (FPR, also called fall-out), False Negative Rate (FNR, also called miss rate), and True Negative Rate (TNR, also called selectivity) are defined as:

$$TPR = \frac{TP}{TP + FN} \quad FPR = \frac{FP}{FP + TN} \quad FNR = \frac{FN}{TP + FN} \quad TNR = \frac{TN}{FP + TN} \quad (3)$$

To our understanding the sum of TPR and FNR must be one, and the sum of FPR and TNR must be one, but there is no reason why the sum of TPR and FPR should be one. The results presented in Fig 6d are coherent with this explanation. Please notice that the values plotted in Fig. 6d of the revised manuscript have been re-calculated to implement the new RRAM relaxation correction programming algorithm (described in Supplementary Material Figs. S1e and f).

12. Minor comments: 1) Several abbreviations repeated multiple times, such as pMUT. 2) Some of the references are not correctly listed, e.g., refs. 40, 49, 60.

Reply 18 Thank you for pointing at these issues which we made sure to correct. We hope now everything is in the correct form.

References

1. Carr, C. & Konishi, M. A circuit for detection of interaural time differences in the brain stem of the barn owl. *J. Neurosci.* **10**, 3227–3246, DOI: <https://doi.org/10.1523/JNEUROSCI.10-10-03227.1990> (1990).
2. Joris, P. X., Smith, P. H. & Yin, T. C. T. Coincidence detection in the auditory system: 50 years after jeffress. *Neuron* **21**, 1235–1238 (1998).
3. Grothe, B. & Park, T. J. Sensitivity to interaural time differences in the medial superior olive of a small mammal, the mexican free-tailed bat. *J. Neurosci.* **18**, 6608–6622, DOI: [10.1523/JNEUROSCI.18-16-06608.1998](https://doi.org/10.1523/JNEUROSCI.18-16-06608.1998) (1998). <https://www.jneurosci.org/content/18/16/6608.full.pdf>.
4. Gutierrez-Galan, D. *et al.* An event-based digital time difference encoder model implementation for neuromorphic systems. *IEEE Transactions on Neural Networks Learn. Syst.* 1–15, DOI: [10.1109/TNNLS.2021.3108047](https://doi.org/10.1109/TNNLS.2021.3108047) (2021).
5. V. Chan, S. L. & van Schaik, A. Aer ear: A matched silicon cochlea pair with address event representation interface. *IEEE Transactions on Circuits Syst. I: Regul. Pap.* **54**, 48–59, DOI: [10.1109/TCSI.2006.887979](https://doi.org/10.1109/TCSI.2006.887979) (2007).
6. Jiménez-Fernández, A. *et al.* A binaural neuromorphic auditory sensor for fpga: A spike signal processing approach. *IEEE Transactions on Neural Networks Learn. Syst.* **28**, 804–818, DOI: [10.1109/TNNLS.2016.2583223](https://doi.org/10.1109/TNNLS.2016.2583223) (2017).
7. Milde, M. B., Bertrand, O. J. N., Ramachandran, H., Egelhaaf, M. & Chicca, E. Spiking Elementary Motion Detector in Neuromorphic Systems. *Neural Comput.* **30**, 2384–2417, DOI: [10.1162/neco_a_01112](https://doi.org/10.1162/neco_a_01112) (2018). https://direct.mit.edu/neco/article-pdf/30/9/2384/1048028/neco_a_01112.pdf.
8. Funabiki, K., Ashida, G. & Konishi, M. Computation of interaural time difference in the owl's coincidence detector neurons. *J. Neurosci.* **31**, 15245–15256, DOI: <https://doi.org/10.1523/JNEUROSCI.2127-11.2011> (2011).
9. Takemura, S.-y. *et al.* A visual motion detection circuit suggested by Drosophila connectomics. *Nature* **500**, 175–181, DOI: <https://doi.org/10.1038/nature12450> (2013). <https://www.nature.com/articles/nature12450>.
10. Maisak, M. S. *et al.* A directional tuning map of Drosophila elementary motion detectors. *Nature* **500**, 212–216, DOI: <https://doi.org/10.1038/nature12320> (2013). <https://www.nature.com/articles/nature12320>.
11. ST-Microelectronics. Ultra-low-power arm cortex-m4 32-bit mcu+fpu, 100dmips, 128kb flash, 40kb sram, analog, aes" stm32l422xx datasheet (2019). [Online] Accessed: January 2022.
12. ST-Microelectronics. Using stm32f4 mcu power modes with best dynamic efficiency (2019). [Online] Accessed: January 2022.
13. Esmanhotto, E. *et al.* High-density 3d monolithically integrated multiple 1t1r multi-level-cell for neural networks. In *2020 IEEE International Electron Devices Meeting (IEDM)*, 36.5.1–36.5.4, DOI: [10.1109/IEDM13553.2020.9372019](https://doi.org/10.1109/IEDM13553.2020.9372019) (2020).
14. Wang, W. *et al.* Learning of spatiotemporal patterns in a spiking neural network with resistive switching synapses. *Sci. Adv.* **4**, eaat4752, DOI: [10.1126/sciadv.aat4752](https://doi.org/10.1126/sciadv.aat4752) (2018). <https://www.science.org/doi/pdf/10.1126/sciadv.aat4752>.
15. Sun, L. *et al.* Synaptic computation enabled by joule heating of single-layered semiconductors for sound localization. *Nano Lett.* **18**, 3229–3234, DOI: [10.1021/acs.nanolett.8b00994](https://doi.org/10.1021/acs.nanolett.8b00994) (2018). PMID: 29668290, <https://doi.org/10.1021/acs.nanolett.8b00994>.
16. Park, P. K. J. *et al.* Fast neuromorphic sound localization for binaural hearing aids. In *2013 35th Annual International Conference of the IEEE Engineering in Medicine and Biology Society (EMBC)*, 5275–5278, DOI: [10.1109/EMBC.2013.6610739](https://doi.org/10.1109/EMBC.2013.6610739) (2013).
17. Faraji, M. M., Shouraki, S. B., Iranmehr, E. & Linares-Barranco, B. Sound source localization in wide-range outdoor environment using distributed sensor network. *IEEE Sensors J.* **20**, 2234–2246, DOI: [10.1109/JSEN.2019.2950447](https://doi.org/10.1109/JSEN.2019.2950447) (2020).
18. Zhen, Z., Shinya, Y. & Shuji, T. Epitaxial pmnn-pzt/si mems ultrasonic rangefinder with 2m range at 1v drive. *Sensors Actuators A: Phys.* **266**, 352–360, DOI: <https://doi.org/10.1016/j.sna.2017.09.058> (2017).
19. Luo, G.-L., Kusano, Y., Roberto, M. N. & Horsley, D. A. High-pressure output 40 khz air-coupled piezoelectric micromachined ultrasonic transducers. In *2019 IEEE 32nd International Conference on Micro Electro Mechanical Systems (MEMS)*, 787–790, DOI: [10.1109/MEMSYS.2019.8870618](https://doi.org/10.1109/MEMSYS.2019.8870618) (2019).

REVIEWERS' COMMENTS

Reviewer #1 (Remarks to the Author):

The paper has improved considerably and it is almost ready for publication after some minor corrections:

- In reply 2, you affirm that FPGAs follow a Von-Neumann architecture. This is not true, since their architecture allows the implementation of any architecture since their born. You can check, just to mention a quick example reference, this sentence: "FPGA's are a reconfigurable systems paradigm that is formulated around the idea of a data stream processor—instead of fetching and processing instructions to operate on data, the data stream processor operates on data directly by means of a multidimensional network of configurable logic blocks (CLBs) connected via programmable interconnects." from this link: <https://community.hitachivantara.com/blogs/hubert-yoshida/2019/02/01/solving-the-von-neumann-bottleneck-with-fpgas>

Nevertheless, this comment is not important in this review, since this sentence from your reply doesn't seem to be present in the current version of the paper.

- In reply 6, if you are not able to measure the real power of your system and you need to estimate the STM32 power consumption, it is fine, but the paper would benefit from the real evidence when offering the power consumption from a real experiment.

Reviewer #2 (Remarks to the Author):

Most of my questions have been addressed. I only have two minor comments:

1. In addition to power consumption, the paper should explicitly mention the comparison of resolution relative to SotA.
2. In page 10, the power of FPGA should be "1.5mW" instead of "1.5uW"?

Response to the reviewers: Neuromorphic object localization using resistive memories and ultrasonic transducers

Once again we would like to thank the reviewers whose comments have continually led us to improve the strength of the manuscript. We are delighted with their positive evaluation of our most recent work as well as our current version of the paper. We have addressed the remaining two questions inline with the reviewer comments below and have made two corresponding improvements to the Discussion section, highlighted using a blue font.

Response to Reviewer 1

The paper has improved considerably and it is almost ready for publication after some minor corrections:

We thank the reviewer for the positive judgement over our work.

In reply 2, you affirm that FPGAs follow a Von-Neumann architecture. This is not true, since their architecture allows the implementation of any architecture since their born. You can check, just to mention a quick example reference, this sentence: "FPGA's are a reconfigurable systems paradigm that is formulated around the idea of a data stream processor—instead of fetching and processing instructions to operate on data, the data stream processor operates on data directly by means of a multidimensional network of configurable logic blocks (CLBs) connected via programmable interconnects." from this link: <https://community.hitachivantara.com/blogs/hubert-yoshida/2019/02/01/solving-the-von-neumann-bottleneck-with-fpgas> Nevertheless, this comment is not important in this review, since this sentence from your reply doesn't seem to be present in the current version of the paper.

Reply 1 We thank the reviewer for this appropriate remark and we apologize for incorrectly stating that FPGAs follow a Von-Neumann architecture. We also thank the reviewer for providing the reference, which allowed us to better focus the issue. We took no action in the paper about this point, because the incorrect statement had not been included in the paper.

In reply 6, if you are not able to measure the real power of your system and you need to estimate the STM32 power consumption, it is fine, but the paper would benefit from the real evidence when offering the power consumption from a real experiment.

Reply 2 We fully agree with the reviewer that a power consumption measurement of the proposed end-to-end system would greatly benefit the paper. The current prototype presented in this paper does not allow us to actually measure the full system power consumption. In this latter respect, in Figure 2(c) of the main text of the revised paper we describe an end-to-end integrated system that will be implemented as part of a new project. This system will enable a direct power consumption measurement and hopefully confirm the power efficiency given by our estimates. We added a line in the main text to clarify this point.

Action 1: We have added a line in the main text, to clarify that the end-to-end experiment to measure power consumption is still to be performed, but it is one of the foreseen extensions of the present work.

Response to Reviewer 2

Most of my questions have been addressed. I only have two minor comments:

We thank the reviewer for going through the manuscript again and for bringing up valuable comments, and in fact we are looking forward to further improving the paper by addressing the comments here below.

In addition to power consumption, the paper should explicitly mention the comparison of resolution relative to SotA.

Reply 3 We thank the reviewer for raising the point of comparing the resolution of the proposed object localization system with the state of the art. In Table 1 we compare the localisation resolution and power per localization with previous contributions, namely two pMUT-based systems for object localization¹ and rangefinding², and a neuromorphic memristor-based system for sound localization³. The system leveraging multiple pMUT devices and classical frame-based signal processing reaches the best localization resolution, but at the expense of orders of magnitude more power per localization operation¹.

Action: We have included considerations on the Angular Resolution in comparison with the literature in the "System assessment" section lines 235-239. We also have added the Table 1 in the Supplementary Information.

	Przybyla [2015] ¹	Chiu [2021] ²	Jin [2014] ⁴	Gao [2022] ³	This work
Application	Object localization 3D	Rangefinding 1D	Sound localization 2D	Sound localization 2D	Object localization 2D
Processing type	Beamforming	TOF estimation	Cross-correlation	Memristor-based Neuromorphic	Memristor-based Neuromorphic
Sensory system	pMUT (2 TX, 7 RX)	pMUT (1 TX, 1 RX)	3 Microphones	2 Microphones	pMUT (1 TX, 2 RX)
Localization precision	0.2° @ 50 cm (angular)	0.63 mm @ 50 cm (distance)	1.45° @ 1 m (angular)	9.0° @ 1 m (angular)	10.0° @ 50 cm (angular)
Power consumption	1.36 mW @ 100 fps	363 μW @ 100 fps	5.63 mW	30.6 μW	81.6 nW @ 100 fps

Table 1. Benchmarking results for the resistive memory based object localization system of this work compared to two pMUT-based systems for object localization and rangefinding, and with a neuromorphic memristor-based system for sound localization.

In page 10, the power of FPGA should be "1.5mW" instead of "1.5uW"?

Reply 4 Thanks for pointing out the typo. We made sure to correct the units in the text.

Action 2: We have changed the units from μW to mW in line 254.

References

1. Przybyla, R. J. *et al.* 3D ultrasonic rangefinder on a chip. *IEEE J. Solid-State Circuits* **50**, 320–334, DOI: [10.1109/JSSC.2014.2364975](https://doi.org/10.1109/JSSC.2014.2364975) (2015).
2. Chiu, Y. *et al.* A novel ultrasonic tof ranging system using AlN based PMUTs. *Micromachines* **12**, DOI: [10.3390/mi12030284](https://doi.org/10.3390/mi12030284) (2021).
3. Gao, B. *et al.* Memristor-based analogue computing for brain-inspired sound localization with in situ training. *Nat. Commun.* **13**, DOI: [10.1038/s41467-022-29712-8](https://doi.org/10.1038/s41467-022-29712-8) (2022).
4. Jin, J. *et al.* Real-time sound localization using generalized cross correlation based on 0.13 CMOS proces. *J. Semicond. Technol. Sci.* **14**, 175–183, DOI: [10.1038/s41467-022-29712-8](https://doi.org/10.1038/s41467-022-29712-8) (2014).